palaeontology, ecology, evolution

ostracoderm, extinction, predation, Silurian, Devonian, ecology

**Author for correspondence:**
Robert S. Sansom
e-mail: robert.sansom@manchester.ac.uk

# Bite marks and predation of fossil jawless fish during the rise of jawed vertebrates

Emma Randle[1,2] and Robert S. Sansom[1]

[1]School of Earth and Environmental Sciences, University of Manchester, Manchester, UK
[2]School of Geography, Earth and Environmental Sciences, University of Birmingham, Birmingham, UK

  ER, 0000-0002-6571-5491; RSS, 0000-0003-1926-2556

Although modern vertebrate diversity is dominated by jawed vertebrates, early vertebrate assemblages were predominantly composed of jawless fishes. Hypotheses for this faunal shift and the Devonian decline of jawless vertebrates include predation and competitive replacement. The nature and prevalence of ecological interactions between jawed and jawless vertebrates are highly relevant to both hypotheses, but direct evidence is limited. Here, we use the occurrence and distribution of bite mark type traces in fossil jawless armoured heterostracans to infer predation interactions. A total of 41 predated specimens are recorded; their prevalence increases through time, reaching a maximum towards the end of the Devonian. The bite mark type traces significantly co-occur with jawed vertebrates, and their distribution through time is correlated with jawed vertebrate diversity patterns, particularly placoderms and sarcopterygians. Environmental and ecological turnover in the Devonian, especially relating to the nekton revolution, have been inferred as causes of the faunal shift from jawless to jawed vertebrates. Here, we provide direct evidence of escalating predation from jawed vertebrates as a potential contributing factor to the demise and extinction of ostracoderms.

## 1. Introduction

The diversification and rise to ecological dominance of jawed vertebrates occurred in the Palaeozoic, approximately coeval with the decline of the ostracoderms (armoured jawless fishes) [1]. This faunal shift is an important episode in the establishment of modern vertebrate diversity, but its circumstances and dynamics are much debated. Raw diversity indices [1,2] clearly show a shift from jawless vertebrate-dominated assemblages in the Silurian to jawed vertebrate-dominated assemblages towards the end of the Devonian. Both groups suffer a mass extinction at the Frasnian/Famennian boundary, from which ostracoderms do not recover. Hypotheses for the gradual decline and extinction of the various ostracoderm clades range from predation or competitive displacement by jawed vertebrates to their limited dispersal capabilities and ability to respond to the environmental change [3–9]. Competitive displacement of ostracoderms by jawed vertebrates is often dismissed due to their presumed ecological dissimilarity; ostracoderms (armoured, jawless stem-gnathostomes) are generally interpreted as being benthic 'mud grubbers', whereas jawed vertebrates are seen as active nektonic predators [8]. In the context of predation, competition or some other scenario, interpretations of the ecological interaction between jawed and jawless fish are central. Correlated patterns of diversity are informative in this context, but hypotheses of causation require direct evidence of ecological interaction. Here, we systematically investigate bite marks and their occurrence as direct evidence for predation of heterostracans (jawless vertebrates) by jawed vertebrates.

Predation has long been recognized in the fossil record and can provide a wealth of information regarding biotic predator–prey interaction in the geological past. Evidence of predation includes trace fossils (bore marks, tooth marks, repair

scars, gnawing and fracture marks), coprolites (fossil faeces), gut contents and even fossilised snapshots of the events themselves [10–12]. Predation of ostracoderms has previously been identified in isolated examples of dermoskeletal bite marks (Devonian heterostracans from the Welsh Borders, Western USA, Baltic and Podolia [4,13–18]). However, the hypothesis of jawless vertebrate extinctions resulting from predation has yet to be quantitatively tested. Here, we present direct evidence of predation in the form of bite marks and scratches in the dermal skeleton of heterostracan ostracoderms. We use the distribution of the traces to test hypotheses of changing patterns of ecological interactions, specifically (i) increasing bite mark prevalence though time, (ii) correlation between bite mark type trace prevalence and jawed vertebrate diversity, and (iii) co-occurrence of bite mark type traces with jawed vertebrates.

## 2. Material and methods

### (a) Identification of predation traces

Following the criteria of Kowalewski [10] and Lebedev *et al.* [14], bite traces were identified by meeting one or more of the following criteria: (i) bite marks are a regular geometric shape, (ii) traces are distributed non-randomly (generally in a linear arrangement reminiscent of tooth arrangement in a jaw), (iii) complementary traces on both sides of the animal, (iv) traces of sublethal damage (signs of repair), (v) gouges and scratch marks with puncture marks, and (vi) obvious deformation of the carapace around the puncture wound (electronic supplementary material, figure S1). Bite occurrences were collected from direct observations of specimens and from the literature and placed in two tiers relating to confidence of identification: tier 1 comprises novel identifications of traces with constrained morphologies that meet multiple bite mark criteria (electronic supplementary material, figure S1) as well as traces previously described in the literature [4,13–18]; traces that met only some of the specific bite mark criteria or occurred in otherwise fragmentary specimens were classified as tier 2 to reflect their more tentative assignment as bite marks. As such, tier 1 traces are those unambiguously interpreted as bite marks following the application of their morphology and preservation to the precise identification criteria (1–6 mentioned earlier), while tier 2 traces were included in a broader dataset retained given their potential evidence of predation. We take two approaches to data analysis: the complete bite mark dataset, or just tier 1 dataset.

### (b) Distribution of predation traces and the associated fauna

Raw numbers of bite occurrences were standardized for sampling effort (number of heterostracan specimens examined per time period). A novel, genus-level occurrence dataset was compiled for jawed vertebrates yielded from heterostracan-bearing horizons (HBHs) from the literature (electronic supplementary material, data). Jawed vertebrate genus diversity was also considered at the level constituent subgroups (placoderms, acanthodians, sarcopterygians, chondrichthyans and actinopterygians) and standardized for sampling using the number of HBHs for each time bin (geological stages). The robustness of correlations was tested using first-order jackknifing (sensitivity following the removal of individual time bins) and first-differences analysis to account for autocorrelation of the time-series data. The robustness of the co-occurrence of jawed vertebrates and heterostracans yielding bite mark type traces was tested using rarefaction analyses at the level of individual HBH (random resampling HBH with 0.66 probability in 100 iterations). Jawed vertebrates were also analysed at the genus level in terms of their frequency of co-occurrence in HBH with bite

mark specimens ($\chi^2$-tests) and their mandible length [2] (Spearman's correlation between the length and frequency of co-occurrence). The dissimilarity between the jawed vertebrate fauna occurring in bite and non-bite HBH was calculated using a permutational MANOVA test (applied using adonis function in the vegan package in R with 10 000 permutations) [19]. For this analysis, horizons were removed if they contained no jawed vertebrate taxa and one horizon was removed if it contained identical jawed vertebrate taxa to another horizon.

## 3. Results

### (a) Bite mark type traces

Bite mark type traces (figure 1; electronic supplementary material, figure S2) were found on a total of 41 heterostracan specimens, of which 29 were classified as tier 1 (10 novel ID and 19 from previous studies) and 12 were classified as tier 2 (electronic supplementary material, table S1). Examples include an articulated dorsal and ventral plate of *Schizosteus asatkini* (figure 1a,b) with complementary bite traces on both dorsal and ventral surfaces, *Tartuosteus maximus* (figure 1d) with a puncture and scratch marks, with signs of repair, and *Psammolepis venyukovi* (figure 1e) with a puncture mark, deformation of the dermal skeleton around the wound and repair.

### (b) Distribution through time

Bite mark type traces occur from the Wenlock (middle Silurian) through the Frasnian (Upper Devonian) spanning the whole of heterostracan evolutionary history. The occurrence of predation traces generally increased through time (figure 2a). The percentage of heterostracan specimens exhibiting bite mark type traces shows a clear increase through time towards the end of the Frasnian extinction event (figure 2b). This relationship is significant for the combined dataset of all bite mark types (Spearman's $\rho = 0.83$, $p = 0.006$, $n = 9$) and for tier 1 bite marks alone (Spearman's $\rho = 0.74$, $p = 0.023$, $n = 9$).

### (c) Relationships between bite mark type traces and jawed vertebrates

Heterostracan specimens were recovered from 137 HBHs from the Wenlock to the Frasnian. Of those HBH, 97 HBHs also yielded specimens of jawed vertebrates (ranging from the Ludlow to the Frasnian although jawed vertebrates are known from earlier non-heterostracan deposits) and 16 yielded bite mark type traces (11 tier 1 type). Within HBH, there was a significant co-occurrence of tier 1 type bite marks type traces with both placoderms ($\chi^2 = 12.2$, $p = 0.0005$, d.f. = 1) and sarcopterygians ($\chi^2 = 4.4$, $p = 0.036$, d.f. = 1), but not other groups of jawed vertebrates. When testing against diversity, there was a significant correlation between tier 1 bite mark type trace prevalence through time and total generic diversity of jawed vertebrates standardized for HBHs (figure 3a,b; Spearman's $\rho = 0.70$, $p = 0.037$, $n = 9$). This relationship was also significant for sarcopterygians ($\rho = 0.82$, $p = 0.007$), but not for other jawed vertebrate subgroups (figure 3c). The same patterns of significant relationships were observed for the complete dataset comprising all bite mark type traces (figure 3c), with the addition of correlation of bite mark type traces with placoderm diversity ($\rho = 0.84$, $p = 0.004$). To account for possible

**Figure 1.** Predation traces in heterostracan dermal skeletons. (*a*–*c*) complementary traces on the dorsal and ventral surface of *S. asatkini* (PIN.220/489), (*d*) puncture mark on ventral plate of *T. maximus* (GIT.116-97) with scratch and dermoskeletal repair, (*e*) predation trace on *P. venyukovi* (GIT.116-212 [14]) with secondary dentine regrowth and repair, (*f*) predation trace in *Rhinopteraspis crouchi* (P.24805) headshield. Scale bars = 10 mm. (*d*–*e*) Baranov/TTÜ GI, Baltic Diversity (CC-BY). (Online version in colour.)

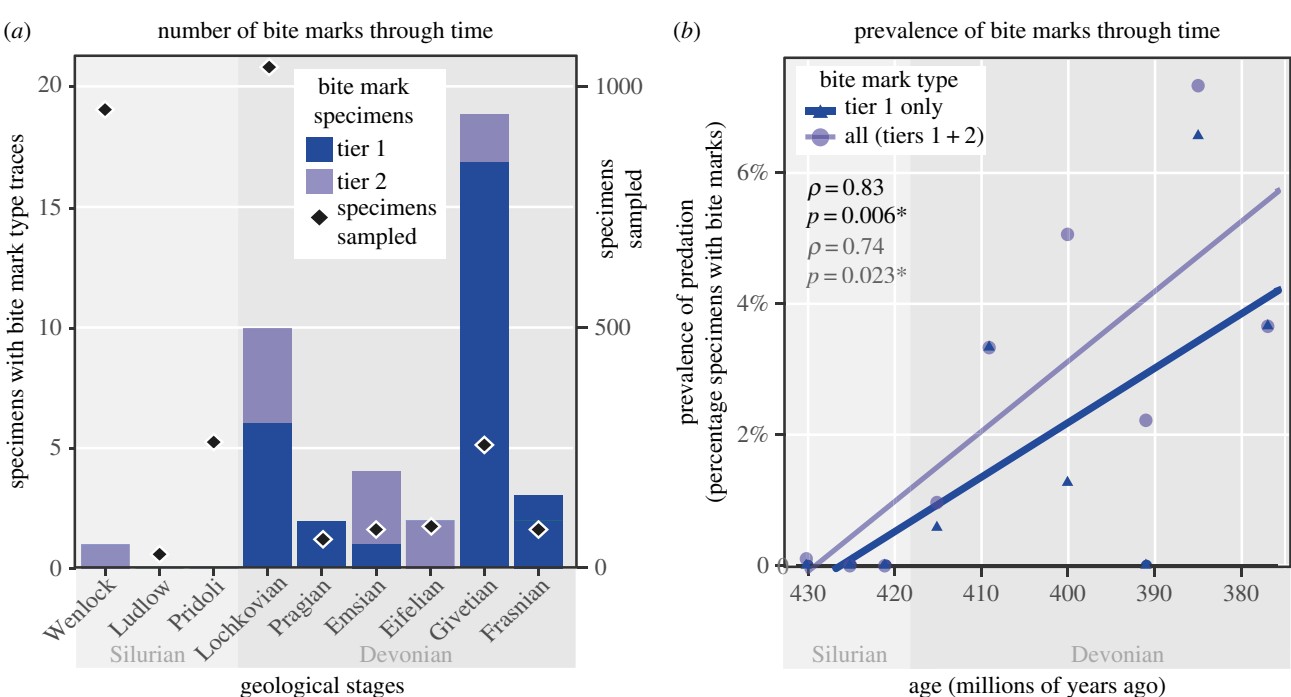

**Figure 2.** Distribution of heterostracan bite mark type traces and sampled specimens through time. (*a*) Numbers of heterostracan specimens yielding bite mark type traces in stage time bins, with numbers of specimens sampled and (*b*) prevalence of bite marks through time normalized by number of specimens sampled per time bin, with Spearman's rank correlation coefficients ($\rho$) and significance values ($p$) for tier 1 only bite marks and complete dataset, respectively. (Online version in colour.)

autocorrelation of time-series data, first-difference tests were applied, and the same correlations were found to be significant; for the tier 1 dataset, bite mark prevalence was correlated with total jawed vertebrate diversity ($\rho = 0.83$, $p = 0.011$) and sarcopterygian diversity ($\rho = 0.74$, $p = 0.035$), while for the complete dataset, bite mark prevalence was

correlated with total jawed vertebrate diversity and sarcopterygian and placoderm diversities ($\rho = 0.92$, $p = 0.002$; $\rho = 0.99$, $p = 5 \times 10^{-7}$; and $\rho = 0.90$, $p = 0.005$, respectively).

The genera of jawed vertebrates most commonly associated with bite marks were identified in terms of their frequency of co-occurrence in HBH: *Panderichthys*, *Grossipterus* (both

Proc. R. Soc. B 286: 20191596

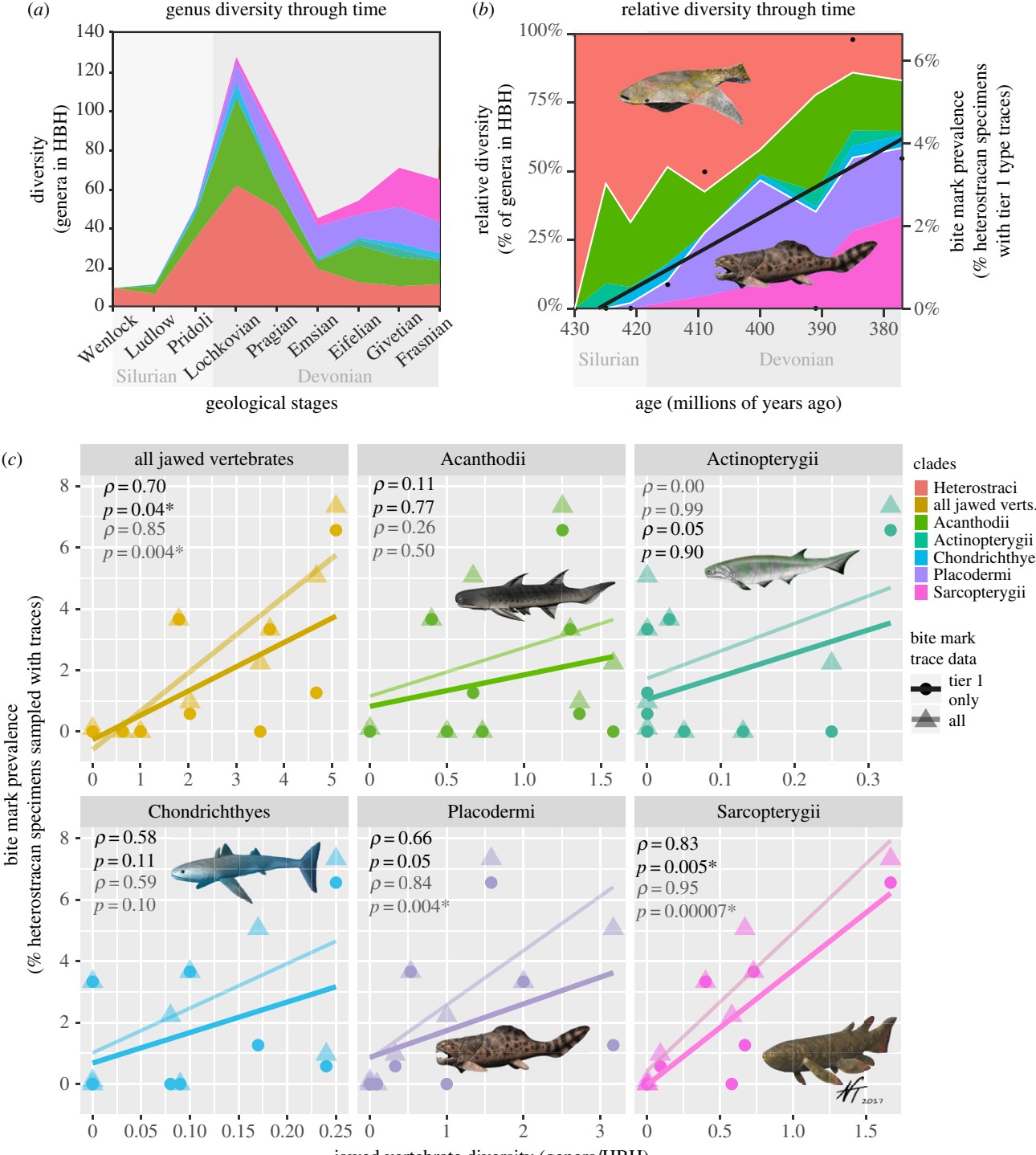

**Figure 3.** Heterostracan and jawed vertebrate generic diversity through time. (a) Number of genera occurring in heterostracan-bearing formations (HBH). (b) Proportions of genera in HBH through time, with bite mark prevalence (tier 1 type) overlain. Colours of groups from (c), with red for Heterostraci. (c) Correlations of diversity of groups of jawed vertebrate with the prevalence of heterostracan bite marks for each geological stage with Spearman's rank correlation coefficients (ρ) and significance values (p). Solid data points, lines and text = tier 1 data (unambiguous predation IDs); lighter data points, lines and text = complete dataset (tiers 1 and 2). Icons for vertebrate groups created by Nobu Tamura (CC-BY). (Online version in colour.)

sarcopterygians), *Livosteus* (a placoderm) and *Nodocosta* (an acanthodian) all occurred in two or more HBHs with tier 1 bite mark heterostracan specimens and did not occur in any HBH without bite mark specimens. These four taxa are all present in the Baltic region only, but patterns appear unrelated to geography: other proximal Baltic sites yield the inverse, and other regions yield high numbers of bite marks (six from the Welsh borderlands). Furthermore, there is no significant relationship between the region and the presence of bite marks type traces (ANOVA $F = 0.48$, $F = 0.61$, $p > 0.9$ for

$n = 138$ HBHs in 31 regions for tier 1 only and complete dataset, respectively). Of the 190 jawed vertebrate genera co-occurring with heterostracans, the mandible length data [2] were available for 53. There was a positive correlation between jawed vertebrate mandible length and the frequency of co-occurrence with tier 1 bite mark specimens (Spearman's $\rho = 0.24$), but this was not significant ($p = 0.087$) (figure 4). The relationship was significant for acanthodians ($\rho = 0.69$, $p = 0.01$), indicating that only larger acanthodians are reliably associated with bite marks. All sarcopterygians for which data are available have

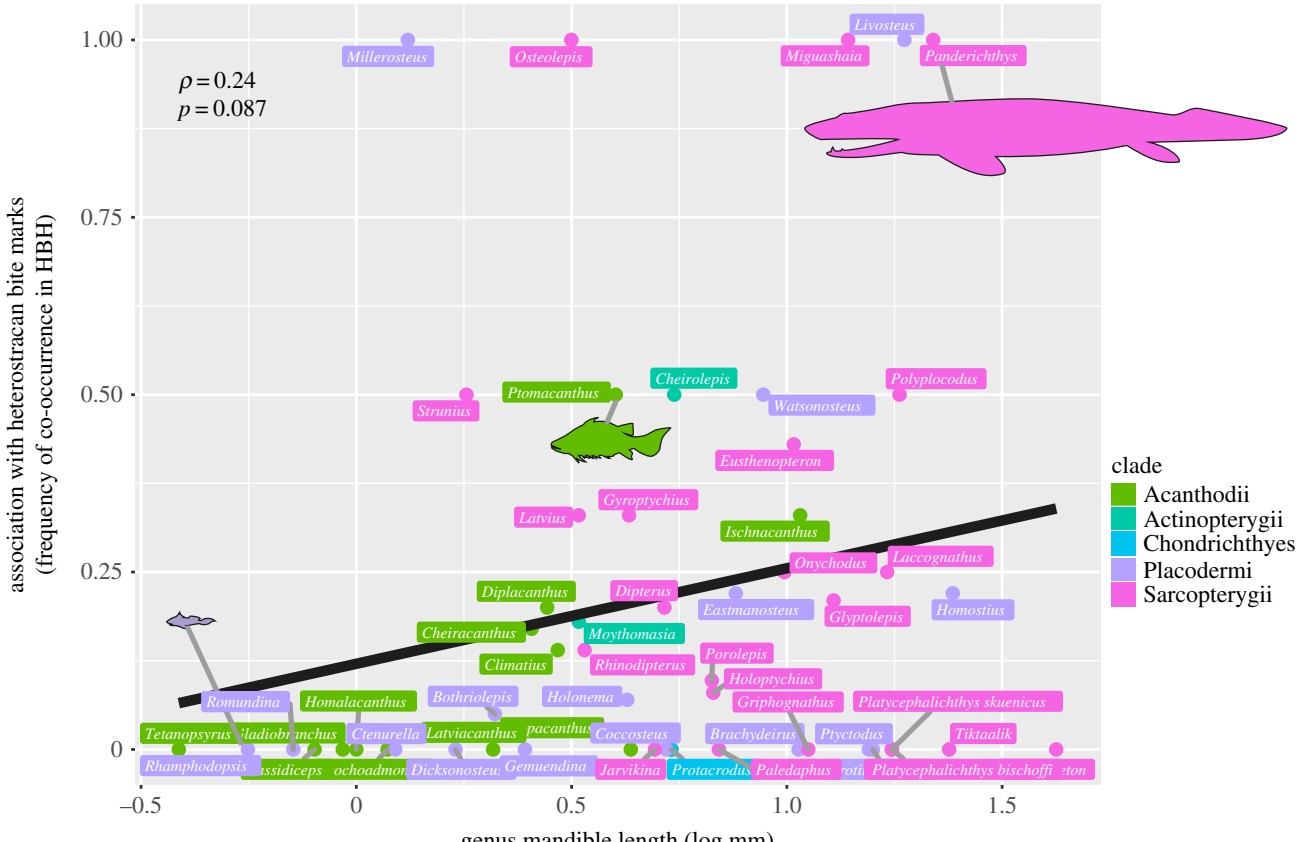

**Figure 4.** Jawed vertebrate mandible length and frequency of co-occurrence with heterostracan specimens exhibiting unambiguous predation marks in heterostracan-bearing formations (HBH). The mandible length data were available for 53 of 190 jawed vertebrate taxa [2]. Data points are colour coded by jawed vertebrate group. The linear correlation is shown, with Spearman's rank correlation coefficient ($\rho = 0.24$, $p = 0.087$). Selected scaled body outlines for *Panderichthys*, *Ptomacanthus* and *Rhamphodopsis*. (Online version in colour.)

generally large jaws (figure 4). No significant difference was found between the faunal composition of jawed vertebrates in HBH that contained specimens with predation traces and horizons that did not (permuted MANOVA $R^2 = 0.062$, f-model = 1.19, $p = 0.07$ for complete bite mark dataset).

## (d) Robustness of patterns

Bite marks were generally rare: 41 of 2846 heterostracan specimens sampled exhibited bite mark type trace (1.4% prevalence), and those bite marks occurred in 14 of 137 heterostracan-bearing formations, across seven of the nine time bins. Rather than the absolute amount of bite marks, the significant patterns observed relate to their relative prevalence through time, their prevalence relative to jawed vertebrate diversity and their co-occurrence with jawed vertebrates in horizons. The application of first-order jackknife tests shows that correlation results are generally robust for the increasing prevalence of bite mark type traces through time (removal of any individual time bin gives correlation of $p < 0.05$ for the complete dataset, and seven of nine time bins for the tier 1 only dataset) and for the correlations of prevalence of bite mark type traces with jawed vertebrate diversity (removal of any time bin gives correlations of $p < 0.05$ for all jawed vertebrates, sarcopterygians and placoderms for the complete dataset, but only sarcopterygians for the tier 1 bite marks). The first-difference analyses were also subjected to first-order jackknife tests, and the originally significant relationships were largely found to be robust (individual removal of eight of nine time bins recovers significant correlations

except the correlation between sarcopterygian diversity and tier 1 bite mark type traces). Power analysis (electronic supplementary material, figure S3) found that Spearman's rank correlation coefficient analyses with the sample size of $n = 9$ have high power for high correlation results only ($\rho > 0.70$); the correlation analyses found to be robust to first-differences analyses and jackknifing fall within this range.

The tests for co-occurrence of bite marks and jawed vertebrates in HBH were subjected to rarefaction analyses (resampling HBH with a probability of ⅔, 100 times). The significant co-occurrence of bite marks and placoderms in HBH was found to be extremely robust (88 and 95 of the 100 resampling iterations recovered chi-square $p < 0.05$ for the complete dataset and tier 1 only dataset, respectively), but the other relationships were less robust (only 25 of 100 iterations were significant for the co-occurrence of sarcopterygians and tier 1 bite mark type traces).

## 4. Discussion

Direct evidence of predation of heterostracan jawless vertebrates is yielded through the occurrence of bite mark type traces (figure 1; electronic supplementary material, figure S2). For example, multiple constrained puncture marks and scratches are observed on the ventral and dorsal surfaces of a *Placosteus* branchial plate, the positions of which correspond on each surface, which would be expected from a crushing or grasping trace maker (electronic supplementary material, table S1). Quantitatively, bite mark traces become increasingly

prevalent through time, peaking towards the end of the Devonian (figure 2a,b). This is the latter end of heterostracan evolutionary history and coincident with the decline of taxonomic diversity of the group (figure 3a,b). In terms of possible predators causing the bite marks, jawed vertebrates were found to significantly co-occur with the specimens yielding bite marks in heterostracan-bearing horizons. Jawed vertebrates, in particular placoderms and sarcopterygians, were significantly associated with bite marks in terms of co-occurrence, and positive correlation of generic diversity with bite mark prevalence (figure 3c). Together, this suggests a possible role of jawed vertebrates as predators of the ostracoderms, particularly sarcopterygians and placoderms.

The individual jawed vertebrate taxa most frequently co-occurring with heterostracan specimens with bite marks include *Panderichthys*, *Livosteus* and *Grossipterus*. All are large, with well-developed jaws and teeth making them the potential predators of jawless vertebrates (figure 5). Previous interpretations of vertebrate bite mark creators based on the morphology included sarcopterygians [14,15] or large acanthodians [14]. This concurs with our quantitative correlations for sarcopterygian diversity. The bite marks analysed here showed variable morphologies, which we have treated by classifying at different tiers of confidence. Around 99% of the total heterostracan specimens analysed exhibited no evidence of predation traces, and this either could indicate that predation of heterostracans was rare or could be due to under sampling given the emphasis on sublethal records (skeletal repair, one of the criteria for bite mark identification [14], was observed in 10 of 41 specimens, while specimens destroyed or consumed through predation were unrecorded). Alternative sources of evidence of ostracoderm predation include stomach contents [20] or preservation of embedded teeth [P Tarrant per coms], but these are difficult to quantify given their exceptional nature and rarity. The low absolute numbers of bite marks in this study (41 of 2846 specimens sampled) make it hard to draw definitive conclusions, but patterns of the increasing prevalence through time and correlations with diversity of jawed vertebrate groups are robust to jackknifing and conservative first-differences tests. Furthermore, both tiers of the trace fossil data (all bite marks, or just tier 1, unambiguous bite marks) show the same overarching results, and their co-occurrence with placoderms is robust to data resampling. Therefore, the combined analyses suggest a possible role of jawed vertebrates in the predation of heterostracans.

During the Silurian and Devonian, possible aquatic predators of jawless vertebrates included jawed vertebrates and eurypterids. Romer's [21] classic hypothesis that the acquisition of the agnathan dermal skeleton was a response to predation pressures from eurypterids was based on similarities between diversity patterns, rather than trace fossil evidence [1,22,23]. Trace fossil evidence has been used to infer eurypterid predation in some instances [13,14], but eurypterid predation has been inferred as more limited or 'low energy' on the basis of functional analyses [7,24–26]. Further investigation into the role of eurypterids as predators may illuminate interesting dynamics between the two groups.

The Devonian epoch was a time of great change in aquatic environments. In the marine realm, the Devonian nekton revolution was occurring, during which many groups took up a free swimming lifestyle, including jawed vertebrates, ammonoids and plankton [26]. This, together with large continental changes (i.e. formation of Euramerica), meant that the

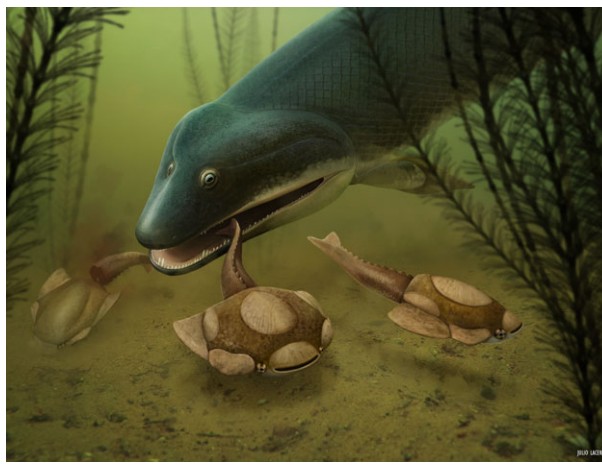

**Figure 5.** Reconstruction of Heterostraci (*Psammolepis*) being predated by a jawed vertebrate (*Panderichthys*) in a Devonian sea scape. Courtesy of Julio Lacerda. (Online version in colour.)

Late Devonian world would have been one of the dramatic environmental change and ecological stresses. The increase in predation traces seen in the Emsian was at a time when vertebrate jaw morphological disparity was at its highest [2] (figure 3). This probably indicates a time of great ecological turnover and could be linked to a shift from benthic to pelagic lifestyles in jawed vertebrates [2]. This, combined with new ecological opportunities relating to the advent of the jaw, meant jawed vertebrates were fast becoming the top predators of the oceans during the Devonian.

Contemporaneous with these biotic changes, ecosystems were also subject to dramatic abiotic changes. Rising sea levels during the Devonian may have adversely affected ostracoderms due to their restriction to freshwater and shallow marine environments [1,9]. This environmental restriction, combined with their limited dispersal capability [27], has been invoked as a causative factor in the decline and eventual extinction of ostracoderms; generic diversity of ostracoderms has been demonstrated to be correlated with relative sea-level changes [1]. The increasing prevalence of bite marks through time identified here indicates that the increasing pressure from predation by jawed vertebrates may have been an additional factor in ostracoderm decline and extinction. In both cases, correlations alone are not sufficient evidence of a causative factor, but in the case of bite marks, we have direct evidence of a biotic interaction, the dynamics of which changed over time.

## 5. Conclusion

Predation traces were identified in a range of heterostracan taxa spanning the majority of their evolutionary history. The increased prevalence of predation was found towards the end of the Devonian; the occurrence of predation traces was found to be significantly correlated with jawed vertebrate occurrence and diversity in HBHs, particularly sarcopterygians and placoderms. It is likely that a combination of environmental and ecological changes, along with the rise to dominance of jawed vertebrates, contributed to the demise of ostracoderms.

Data accessibility. Fossil occurrence data are available in supplementary information from the Dryad Digital Repository: https://dx.doi.org/10.5061/dryad.f32p5g7 [28].

**Authors' contributions.** E.R. and R.S.S. conceived the project and wrote the manuscript. E.R. collected the data, which were analysed by R.S.S. and E.R. Both E.R. and R.S.S. wrote the manuscript.

**Competing interests.** We have no competing interests.

**Funding.** E.R. was supported by a NERC studentship (grant no. NE/K500859/1), a Synthesys grant (grant no. SE-TAF-3875), a PalAss Small (grant no. PA-SW201602) and R.S.S. supported by BBSRC (grant no. BB/N015827/1).

**Acknowledgements.** We thank staff at all museums for access to specimens, including Dr Olga Afanassieva and Dr Larissa Novitskaya (Palaeontological Institute, Moscow), Dr Tiiu Märss and Ursula Toom (Geological Institute, Tallinn), Jonathan Clatworthy and Dr Ivan Sansom (Lapworth Museum of Geology, University of Birmingham), Emma Bernard (Natural History Museum, London), William Simpson and Akiko Shinya (Field Museum, Chicago), Kiaeron Sheppard and Margaret Curry (Canadian Museum of Nature) and Dr Lars Werdelin and Jonas Hagström (Naturhistoriska Riksmuseet, Stockholm), Dr Stig Walsh (National Museums Scotland), John Bruner, Dr. Alison Murray and Dr Mark Wilson (University of Alberta, Edmonton). We also thank Dr Vadim Glinskiy and Tormi Tuuling for their help in acquiring papers, and Dr Zerina Johanson, Dr Joseph Keating, Dr Marco Castiello, Dr Richard Dearden, Dr Martin Brazeau, Peter Tarrant and David Marshall for constructive discussions, acquisition of literature and feedback. Palaeoart reconstruction in figure 5 created by Julio Lacerda. We would also like to thank Nobu Tamura for use of his artwork in our figures. We thank the four anonymous reviewers and editorial board for helpful feedback that improved the manuscript.

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
