## [Reviewer comments · Proceedings of the Royal Society B: Biological Sciences]

Review History

RSPB-2019-0585.R0 (Original submission)

Review form: Reviewer 1

Recommendation

Accept with minor revision (please list in comments)

Scientific importance: Is the manuscript an original and important contribution to its field?

Good

General interest: Is the paper of sufficient general interest?

Excellent

Quality of the paper: Is the overall quality of the paper suitable?

Good

Is the length of the paper justified?

Yes

Should the paper be seen by a specialist statistical reviewer?

No

Do you have any concerns about statistical analyses in this paper? If so, please specify them explicitly in your report.

No

It is a condition of publication that authors make their supporting data, code and materials available - either as supplementary material or hosted in an external repository. Please rate, if applicable, the supporting data on the following criteria.

Is it accessible?

Yes

Is it clear?

Yes

Is it adequate?

Yes

Do you have any ethical concerns with this paper?

No

Comments to the Author

The decline of armored jawless fishes and the associated rise of jawed fishes is one of the most conspicuous features of the early vertebrate fossil record, and set the stage for the subsequent 400 million years of evolution in backboned animal evolution. The causes of this stark change in faunas during the Devonian has been much debated, largely from the perspective of diversity metrics (i.e. taxonomic richness and morphological disparity). At best, such approaches offer circumstantial evidence for evaluating a range of competing hypotheses. One popular explanation for the decline of jawless fishes in the face of gnathostome diversification invokes direct interaction between these groups, in the form of increased predation pressure by jawed vertebrates. Past evidence for this has been, to put it generously, anecdotal and diffuse.

The authors provide a comprehensive survey of a particularly diverse group of jawless vertebrates (heterostracans) over their Silurian-Devonian record. They report direct evidence of (failed) predation in the form of bite marks on the armor plates of these jawless fishes, probably (but not certainly) left by jawed fishes. The authors clearly demonstrate two important relationships. First, that bite marks increase over the history of heterostracans, and second, that the occurrence of bite marks correlates with the co-occurrence of certain large groups of jawed fish predators. While this study does not provide a definitive answer to the extinction of armored jawless fishes (and the authors acknowledge this fully), it nevertheless provides an important line of evidence that has previously received shockingly little attention. I imagine that the plots presented here are the kinds of things that are likely to appear in lectures in vertebrate paleontology classes, and it would not be a stretch to imagine them finding a place in textbooks.

I have made a few suggestions in the text, but these largely concern wording and typographical errors. The most substantive changes I recommend in my in-text suggestions are:

(1) The use of non-parametric rank-order tests for establishing patterns of covariation (there is no reason to assume that the correlations here are linear). I am reasonably sure that the results presented here will be robust to this alternative approach.

(2) The use of taxonomically and stratigraphically appropriate group icons in Figure 3. The sarcopt icon is in fact an actinopt (Cheirolepis), while the use of a hammerhead shark and a paddlefish for chondrichthyans and actinopt is about as phylogenetically and stratigraphically incongruous as alternatives like a torpedo ray and seahorse.

There are a few additional things I would ask the authors to consider, and which would only really require the addition of a line or two to the text. It would be very easy to add something more substantive, since these data are available 'off the shelf'. I think this would add considerable value to this paper:

(1) At least an consideration of body size. One of the most striking things (to my mind, anyhow) is the very large size of some of the youngest heterostracans. These data are readily available, in the supplement of Sallan & Galimberti (2015, Science). There are obviously many reasons why body size might increase within a group over time, but it seems like this is something worth mentioning as a possible (but untested) response to predation.

(2) Related to the above, it would be interesting to consider whether gape size increased in gnathostomes over this same interval. The data are available in the supplement to Anderson et al. (2011, Nature).

Review form: Reviewer 2

Recommendation

Reject – article is not of sufficient interest (we will consider a transfer to another journal)

Scientific importance: Is the manuscript an original and important contribution to its field?

Acceptable

General interest: Is the paper of sufficient general interest?

Marginal

Quality of the paper: Is the overall quality of the paper suitable?

Acceptable

Is the length of the paper justified?

Yes

Should the paper be seen by a specialist statistical reviewer?

Yes

Do you have any concerns about statistical analyses in this paper? If so, please specify them explicitly in your report.

Yes

It is a condition of publication that authors make their supporting data, code and materials available - either as supplementary material or hosted in an external repository. Please rate, if applicable, the supporting data on the following criteria.

Is it accessible?

Yes

Is it clear?

Yes

Is it adequate?

Yes

Do you have any ethical concerns with this paper?

No

Comments to the Author

The authors present bite mark data on jawless vertebrates from the Silurian and Devonian periods to address the question of predation as a potential driver for the reduction in jawless vertebrate diversity towards the end Devonian. They examine the number of specimens showing “bite marks” relative to the total number of specimens examined across time bins and see an increase in bite marks towards the end Frasnian. They use these results to argue that increased predation may be a cause of jawless vertebrate diversity reduction.

I appreciate the attempt to use an oft-overlooked form of trace fossil data (predation marks) to address broader questions in paleoecology. However, I find the data presented too sparse to make the conclusions that the authors are trying to make. They looked at over 2900 specimens and found, at most, 41 bite mark cases. Meaning they only found predation evidence on less than 2% of the specimens examined. Furthermore, some of these are classified as “tier 2” meaning they don’t actually fit the criteria laid out in the methods. The authors don’t give enough detail about what constitutes a “tier 2” bite mark, making it unclear if these really are evidence of predation. Two of the tier-2 instances are noted to not come from localities that have associated gnathostome fossils, meaning there is no evidence of jawed predators. At the very least, these two specimens should be removed from the analysis. I would suggest only including tier-1 bite marks (note, the authors do show tier 1 only results as well), but this reduces the amount of actual data to less than 1% of the specimens examined.

Amount of data aside, the pattern presented in figure 2, meant to show increasing bite marks through time, is not convincing. The trend is partly driven by the first three bins showing essentially no bite marks across the largest number of specimens examined. The Wenlock has almost 1000 specimens checked by itself (a third of the total data set) and only a single, tier-2 bite mark. These bins are certainly going to exert an inordinate amount of influence on the regression, but it is unclear whether there even were gnathostome predators around. The one bite mark found in the Wenlock is tier-2, and I assume one of the two that isn’t associated with gnathostomes. A lack of predators in these early bins is likely a real signal, but if the goal of this study is to assess whether predation itself increased, only localities that include both jawless and jawed vertebrates should be used. This means removing all of the jawless specimens from their count of 2900 that appear without co-occurring jawed forms.

Figure 2 also shows that the signal in the later Devonian seems to be strongly influenced by one time bin, the Givetian. It is unclear how much isolated localities may be effecting these numbers, but that is another aspect of the data that must be examined.

There are a series of steps the authors must take to alleviate these issues:

- 1) Remove tier 2 data from the analysis, or at least the two examples that are not associated with any gnathostome fossils. In general, if the interest is in the trend of predation, no jawless specimens from localities that lack gnathostomes should be used
- 2) A statistical power analysis is vital to convince the reader that there is enough data to actually

support the proposed trend.

3) The authors should perform a jackknifing procedure to assess the influence of exceptional fossil assemblages that may be biasing the results; particularly in the Givetian, which appears to have more bite mark evidence than any other bin including the later Frasnian.

These steps are vital to determining whether the data is sufficient to actually support the proposed trend.

A couple other issues:

Discussion, Pg. 6, Ln. 167-172: It is noted that the most likely predators of jawless fish at the time would be acanthodians, sarcopterygians and actinopterygians. However, the data shows that only the placoderms and sarcopterygians were significantly associated with bite marks. This further undercuts the conclusions as the placoderms that seemingly drive much of the co-occurrence data are considered NOT to be viable predators.

Figure 3: The Y-axis for part C are labeled as "number of predation traces". However, these axis have values of less than 1. Presumably these are standardized values? That should be made explicit in the label.

Review form: Reviewer 3 (Lauren Sallan)

Recommendation

Major revision is needed (please make suggestions in comments)

Scientific importance: Is the manuscript an original and important contribution to its field?

Excellent

General interest: Is the paper of sufficient general interest?

Excellent

Quality of the paper: Is the overall quality of the paper suitable?

Good

Is the length of the paper justified?

Yes

Should the paper be seen by a specialist statistical reviewer?

No

Do you have any concerns about statistical analyses in this paper? If so, please specify them explicitly in your report.

Yes

It is a condition of publication that authors make their supporting data, code and materials available - either as supplementary material or hosted in an external repository. Please rate, if applicable, the supporting data on the following criteria.

Is it accessible?

Yes

Is it clear?

Yes

Is it adequate?

Yes

Do you have any ethical concerns with this paper?

No

Comments to the Author

Randle and Sansom have produced a very interesting study of the incidence of predation on jawless fishes by their jawed cousins. Direct evidence for this kind of predation has not yet been fully documented outside scattered observations, or quantified, preventing further discussion and explicit tests of the role of jawed vertebrates in jawless fish decline and extinction. The authors have done a thorough job of combing museum collections and identifying predation traces in one group of armored fishes, the heterostracans, according the standards of the field, and have made a solid attempt to connect the incidence of predation damage with the presence of different jawed predators. This is an excellent and much needed effort, but I have some concerns about whether the statistical tests and comparisons used are the most appropriate or necessary. Thus, I recommend something like moderate revisions to the methods, even though that's not an option on the pull-down menu. However, I don't anticipate this will take long or require a large effort on the author's part, and I eagerly look forward to seeing this important study published!

Methods and Use of Statistics:

I am not sure all the statistical tests are necessary or appropriate given the clarity of the general pattern. That said, some of the variables need to be better standardized and aligned with the hypotheses. For example, It seems like a bit of an "apples to oranges" comparison to use absolute number of predation traces (which may be made by a specialized single predator) and genus diversity (which includes all guilds, even those which cannot eat jawless fishes like *Acanthodes* or *Bothriolepis*) in the tests in Fig. 3. That is, it's not clear what the relationship should be, or what would drive a significant correlation. It might be better to test something like the association between numbers with bite marks/percentage with bite marks and number of jawed fish specimens from the same site/percentage of taxa which are jawed. That way the variables are in the same general class.

You could also compare number of distinct types of predation traces (punctures or damage) with genus diversity, because then it would be number of kinds of attackers vs. number of kinds of attacks. But in that case, the diversity of jawed fishes should be limited to those with biting predation and/or the kinds of teeth which can produce observed damage (e.g. taxa with coronoid fangs rather than durophages).

In all correlation tests of time series, first differencing should be used rather than the full value to prevent autocorrelation (see Sallan et al. 2011). Then you are explicitly testing the correlation between the direction of change, and reduce any effects of large increases in jawed fish diversity and abundance. Is the positive correlation with sarcopts and placoderms driven by greater sample size for those groups in an interval of increased gnathostome dominance?

It is not clear that the Chi Square tests of predation trace occurrence and gnathostome presence add anything given that all with tier 1 predation traces have jawed fishes, that involving tier 2 removes significance, and the small sample sizes relative to the number of sites examines. It's probably enough to just note the general pattern.

An alternative to the regression and chi-square tests would be to test if there are any predatory genera/families/clades/predation categories more closely associated with the 11/16 sites with bite marks vs. the 127/122 without. Some likely taxa are noted in the discussion (lines 165-166) but this could be tested explicitly, perhaps revealing other associations. Appropriate tests would include ANOSIM and SIMPER (see the book *Paleontological Data Analysis* by Hammer and Harper). Your supplementary data sheet "Jawed Vertebrate HBH" is almost in the right format for doing this at the genus or group level. It might just require some rearrangement of the rows into groups with extinction traces and without overall or by stage.

Discussion:

Is it possible Tier 2 traces were produced by something else, such as arthropods or mollusks given the absence of jawed fishes at two of the sites? This possibility is raised briefly in the discussion but not made explicit. How do these traces compare to those assumed to be eurypterid as noted in lines 188-189?

The discussion mentions that most of the traces are likely sublethal. It would be good to know explicitly about how many traces were fatal vs. non-fatal, given evidence for regrowth.

Another good paper dealing with change in gnathostome traces, and identification of fatalities, through time is Salamon et al. *Paleobiology* 2013

Decision letter (RSPB-2019-0585.R0)

08-Apr-2019

Dear Dr Sansom:

I am writing to inform you that your manuscript RSPB-2019-0585 entitled "Bite marks and predation of fossil jawless fish during the rise of jawed vertebrates" has, in its current form, been rejected for publication in *Proceedings B*.

This action has been taken on the advice of referees, who have recommended that substantial revisions are necessary. With this in mind we would be happy to consider a resubmission, provided the comments of the referees are fully addressed. However please note that this is not a provisional acceptance. Indeed, while the reviewers and editor found value in the paper, they had lengthy and detailed constructive critiques that must be attentively addressed if you choose the route of resubmission, and even then rejection may be an outcome.

In your revision process, please take a second look at how open your science is; our policy is that all data involved with the study should be made openly accessible-- see: <https://royalsociety.org/journals/ethics-policies/data-sharing-mining/>
Insufficient sharing of data can delay or even cause rejection of a paper.

Sincerely,
Professor John R. Hutchinson, Editor
Proceedings B
mailto: proceedingsb@royalsociety.org

Associate Editor
Board Member: 1
Comments to Author:

Randle and Sansom provide a comprehensive and impressive survey of predation traces on heterostracans, a group of jawless fishes, from the Silurian to Devonian. The authors found that the proportion of bite marks increased towards the end of the Frasnian, which they suggested was linked to the increasing prevalence of jawed fishes. Therefore, the decline of jawless fishes was potentially driven by predation from their jawed cousins. The paper is well written, the presentation is straightforward, and I commend the authors on what was clearly a large amount of work. All three reviewers, however, raised a number of issues that need to be addressed prior to publication. I also detail some additional comments below, which may further improve this already-excellent contribution.

The statistical tests were mentioned as needing improvement by all three reviewers, and I concur. For example, the authors need to (at least) test whether their time series correlations are stationary (e.g., using the Augmented Dickey-Fuller Test). If their data are not stationary, first differencing would be appropriate, as mentioned by Reviewer 3. Correlation coefficients (in addition to p-values) need to be provided in the text so that readers can gauge the strength of reported relationships. Finally, Reviewer 2 provided good suggestions to examine the robustness of patterns, such as jackknife and power analyses.

The authors correlated jawed vertebrate diversity against heterostracan bite prevalence through time, but I wondered how this diversity was calculated? That is, did the authors use raw diversity estimates, or was diversity subsampled/standardized? If only the former, I would suggest examining patterns using the latter, since this would help isolate whether diversity is simply a correlate of available sedimentary packages (for example).

The authors discussed tier 1 versus tier 2 bite marks, but it was unclear from reading the text what constituted a tier 2 bite mark, and how tier 2 bites were distinguished from tier 1 bite marks.

This should be further explicated. Moreover, Reviewer #3 raised the question of why all jawed vertebrate guilds are included in the diversity analyses, when some of these groups (e.g., Acanthodes) cannot eat jawless fishes.

Fig S3 doesn't seem to be referenced in the main text, and I wasn't clear what this figure depicted. Did authors examine predation intensity in jawed vertebrates within HBFs, in addition to predation intensity in jawless fishes? If so, the total number of specimens examined should be included here, as was done in Fig 2. Moreover, what do patterns look like through time? Does the prevalence of bite marks for jawed vertebrates also increase? If yes, would this affect the authors' conclusions?

I was concerned that one of the strongest relationships between predation frequency and jawed vertebrates was for the placoderm *Asterolepis*, which the authors indicated would have been unlikely to predate jawless fishes (lines 63-68). I would suggest that all taxa that are not candidate predators be removed from all analyses and patterns re-analyzed.

The analyses suggested by Reviewer 1, which focus on examination of gape size and body size in relation to predation intensity, would broaden the reach of this contribution even further.

Line 15: this is the first time heterostracans are mentioned, so please define.

Lines 24-26: consider toning down the confidence level for the last sentence of the abstract.

Line 46: remove 'a' prior to benthic mud grubbers

Line 62: remove 's' on vertebrates

Line 166: need a space between parentheses

Line 184: extra space between 'skeleton' and 'was'

Line 188: comma before 'but eurypterid...'

Fig 2, part b: a blue dot appears at -430 (x-axis) above the figure legend. Is this an occurrence?

Fig 3, part c: I am assuming the y-axis values are proportions? If so, this should be noted.

Reviewer(s)' Comments to Author:

Referee: 1

Comments to the Author(s)

The decline of armored jawless fishes and the associated rise of jawed fishes is one of the most conspicuous features of the early vertebrate fossil record, and set the stage for the subsequent 400 million years of evolution in backboned animal evolution. The causes of this stark change in faunas during the Devonian has been much debated, largely from the perspective of diversity metrics (i.e. taxonomic richness and morphological disparity). At best, such approaches offer circumstantial evidence for evaluating a range of competing hypotheses. One popular explanation for the decline of jawless fishes in the face of gnathostome diversification invokes direct interaction between these groups, in the form of increased predation pressure by jawed vertebrates. Past evidence for this has been, to put it generously, anecdotal and diffuse.

The authors provide a comprehensive survey of a particularly diverse group of jawless vertebrates (heterostracans) over their Silurian-Devonian record. They report direct evidence of (failed) predation in the form of bite marks on the armor plates of these jawless fishes, probably (but not certainly) left by jawed fishes. The authors clearly demonstrate two important relationships. First, that bite marks increase over the history of heterostracans, and second, that the occurrence of bite marks correlates with the co-occurrence of certain large groups of jawed fish predators. While this study does not provide a definitive answer to the extinction of armored jawless fishes (and the authors acknowledge this fully), it nevertheless provides an important line of evidence that has previously received shockingly little attention. I imagine that the plots presented here are the kinds of things that are likely to appear in lectures in vertebrate paleontology classes, and it would not be a stretch to imagine them finding a place in textbooks.

I have made a few suggestions in the text, but these largely concern wording and typographical errors. The most substantive changes I recommend in my in-text suggestions are:

- (1) The use of non-parametric rank-order tests for establishing patterns of covariation (there is no reason to assume that the correlations here are linear). I am reasonably sure that the results presented here will be robust to this alternative approach.
- (2) The use of taxonomically and stratigraphically appropriate group icons in Figure 3. The sarcopt icon is in fact an actinopt (Cheirolepis), while the use of a hammerhead shark and a paddlefish for chondrichthyans and actinopt is about as phylogenetically and stratigraphically incongruous as alternatives like a torpedo ray and seahorse.

There are a few additional things I would ask the authors to consider, and which would only really require the addition of a line or two to the text. It would be very easy to add something more substantive, since these data are available 'off the shelf'. I think this would add considerable value to this paper:

- (1) At least an consideration of body size. One of the most striking things (to my mind, anyhow) is the very large size of some of the youngest heterostracans. These data are readily available, in the supplement of Sallan & Galimberti (2015, Science). There are obviously many reasons why body size might increase within a group over time, but it seems like this is something worth mentioning as a possible (but untested) response to predation.
- (2) Related to the above, it would be interesting to consider whether gape size increased in gnathostomes over this same interval. The data are available in the supplement to Anderson et al. (2011, Nature).

Referee: 2

Comments to the Author(s)

The authors present bite mark data on jawless vertebrates from the Silurian and Devonian periods to address the question of predation as a potential driver for the reduction in jawless vertebrate diversity towards the end Devonian. They examine the number of specimens showing "bite marks" relative to the total number of specimens examined across time bins and see an increase in bite marks towards the end Frasnian. They use these results to argue that increased predation may be a cause of jawless vertebrate diversity reduction.

I appreciate the attempt to use an oft-overlooked form of trace fossil data (predation marks) to address broader questions in paleoecology. However, I find the data presented too sparse to

make the conclusions that the authors are trying to make. They looked at over 2900 specimens and found, at most, 41 bite mark cases. Meaning they only found predation evidence on less than 2% of the specimens examined. Furthermore, some of these are classified as “tier 2” meaning they don’t actually fit the criteria laid out in the methods. The authors don’t give enough detail about what constitutes a “tier 2” bite mark, making it unclear if these really are evidence of predation. Two of the tier-2 instances are noted to not come from localities that have associated gnathostome fossils, meaning there is no evidence of jawed predators. At the very least, these two specimens should be removed from the analysis. I would suggest only including tier-1 bite marks (note, the authors do show tier 1 only results as well), but this reduces the amount of actual data to less than 1% of the specimens examined.

Amount of data aside, the pattern presented in figure 2, meant to show increasing bite marks through time, is not convincing. The trend is partly driven by the first three bins showing essentially no bite marks across the largest number of specimens examined. The Wenlock has almost 1000 specimens checked by itself (a third of the total data set) and only a single, tier-2 bite mark. These bins are certainly going to exert an inordinate amount of influence on the regression, but it is unclear whether there even were gnathostome predators around. The one bite mark found in the Wenlock is tier-2, and I assume one of the two that isn’t associated with gnathostomes. A lack of predators in these early bins is likely a real signal, but if the goal of this study is to assess whether predation itself increased, only localities that include both jawless and jawed vertebrates should be used. This means removing all of the jawless specimens from their count of 2900 that appear without co-occurring jawed forms.

Figure 2 also shows that the signal in the later Devonian seems to be strongly influenced by one time bin, the Givetian. It is unclear how much isolated localities may be effecting these numbers, but that is another aspect of the data that must be examined.

There are a series of steps the authors must take to alleviate these issues:

- 1) Remove tier 2 data from the analysis, or at least the two examples that are not associated with any gnathostome fossils. In general, if the interest is in the trend of predation, no jawless specimens from localities that lack gnathostomes should be used
- 2) A statistical power analysis is vital to convince the reader that there is enough data to actually support the proposed trend.
- 3) The authors should perform a jackknifing procedure to assess the influence of exceptional fossil assemblages that may be biasing the results; particularly in the Givetian, which appears to have more bite mark evidence than any other bin including the later Frasnian.

These steps are vital to determining whether the data is sufficient to actually support the proposed trend.

A couple other issues:

Discussion, Pg. 6, Ln. 167-172: It is noted that the most likely predators of jawless fish at the time would be acanthodians, sarcopterygians and actinopterygians. However, the data shows that only the placoderms and sarcopterygians were significantly associated with bite marks. This further undercuts the conclusions as the placoderms that seemingly drive much of the co-occurrence data are considered NOT to be viable predators.

Figure 3: The Y-axis for part C are labeled as “number of predation traces”. However, these axis have values of less than 1. Presumably these are standardized values? That should be made explicit in the label.

Referee: 3

Comments to the Author(s)

Randle and Sansom have produced a very interesting study of the incidence of predation on jawless fishes by their jawed cousins. Direct evidence for this kind of predation has not yet been fully documented outside scattered observations, or quantified, preventing further discussion and explicit tests of the role of jawed vertebrates in jawless fish decline and extinction. The authors have done a thorough job of combing museum collections and identifying predation traces in one group of armored fishes, the heterostracans, according the standards of the field, and have made a solid attempt to connect the incidence of predation damage with the presence of different jawed predators. This is an excellent and much needed effort, but I have some concerns about whether the statistical tests and comparisons used are the most appropriate or necessary. Thus, I recommend something like moderate revisions to the methods, even though that's not an option on the pull-down menu. However, I don't anticipate this will take long or require a large effort on the author's part, and I eagerly look forward to seeing this important study published!

Methods and Use of Statistics:

I am not sure all the statistical tests are necessary or appropriate given the clarity of the general pattern. That said, some of the variables need to be better standardized and aligned with the hypotheses. For example, It seems like a bit of an "apples to oranges" comparison to use absolute number of predation traces (which may be made by a specialized single predator) and genus diversity (which includes all guilds, even those which cannot eat jawless fishes like *Acanthodes* or *Bothriolepis*) in the tests in Fig. 3. That is, it's not clear what the relationship should be, or what would drive a significant correlation. It might be better to test something like the association between numbers with bite marks/percentage with bite marks and number of jawed fish specimens from the same site/percentage of taxa which are jawed. That way the variables are in the same general class.

You could also compare number of distinct types of predation traces (punctures or damage) with genus diversity, because then it would be number of kinds of attackers vs. number of kinds of attacks. But in that case, the diversity of jawed fishes should be limited to those with biting predation and/or the kinds of teeth which can produce observed damage (e.g. taxa with coronoid fangs rather than durophages).

In all correlation tests of time series, first differencing should be used rather than the full value to prevent autocorrelation (see Sallan et al. 2011). Then you are explicitly testing the correlation between the direction of change, and reduce any effects of large increases in jawed fish diversity and abundance. Is the positive correlation with sarcopts and placoderms driven by greater sample size for those groups in an interval of increased gnathostome dominance?

It is not clear that the Chi Square tests of predation trace occurrence and gnathostome presence add anything given that all with tier 1 predation traces have jawed fishes, that involving tier 2 removes significance, and the small sample sizes relative to the number of sites examines. It's probably enough to just note the general pattern.

An alternative to the regression and chi-square tests would be to test if there are any predatory genera/families/clades/predation categories more closely associated with the 11/16 sites with bite marks vs. the 127/122 without. Some likely taxa are noted in the discussion (lines 165-166) but this could be tested explicitly, perhaps revealing other associations. Appropriate tests would include ANOSIM and SIMPER (see the book *Paleontological Data Analysis* by Hammer and

Harper). Your supplementary data sheet "Jawed Vertebrate HBH" is almost in the right format for doing this at the genus or group level. It might just require some rearrangement of the rows into groups with extinction traces and without overall or by stage.

Discussion:

Is it possible Tier 2 traces were produced by something else, such as arthropods or mollusks given the absence of jawed fishes at two of the sites? This possibility is raised briefly in the discussion but not made explicit. How do these traces compare to those assumed to be eurypterid as noted in lines 188-189?

The discussion mentions that most of the traces are likely sublethal. It would be good to know explicitly about how many traces were fatal vs. non-fatal, given evidence for regrowth.

Another good paper dealing with change in gnathostome traces, and identification of fatalities, through time is Salamon et al. Paleobiology

Author's Response to Decision Letter for (RSPB-2019-0585.R0)

See Appendix A.

RSPB-2019-1596.R0

Review form: Reviewer 1

Recommendation

Accept with minor revision (please list in comments)

Scientific importance: Is the manuscript an original and important contribution to its field?

Acceptable

General interest: Is the paper of sufficient general interest?

Good

Quality of the paper: Is the overall quality of the paper suitable?

Acceptable

Is the length of the paper justified?

Yes

Should the paper be seen by a specialist statistical reviewer?

No

Do you have any concerns about statistical analyses in this paper? If so, please specify them explicitly in your report.

No

It is a condition of publication that authors make their supporting data, code and materials available - either as supplementary material or hosted in an external repository. Please rate, if applicable, the supporting data on the following criteria.

Is it accessible?

Yes

Is it clear?

Yes

Is it adequate?

No

Do you have any ethical concerns with this paper?

No

Comments to the Author

This contribution examines the frequency of inferred predation traces on heterostracan (armored jawless fish bones) over the latest Silurian and Devonian, with a goal of establishing whether such marks--and by extension, the intensity of predation pressure exerted by jawed fishes--increased over this interval. The authors finds an increase in bite mark frequency over time, and concludes that increased predation remains as a possible driver of extinction in 'ostracoderms'.

While I think that this study delivers some much-needed data bearing on a classic 'just-so' story in vertebrate paleontology, I think it would benefit from a bit more reflection the (i) limitations of the data at hand, (ii) the degree to which any result might constrain our understanding of 'ostracoderm' extinction, and (iii) slightly more complete supplementary material.

Specific comments are on a marked copy of the MS, and more general questions/comments are here.

1) More information should be provided to readers to determine if there is a site-specific pattern. In terms of overall counts of tier 1 marks within individual horizons, there are Givetian sites with 10 and 5 such traces, and one Lochkovian one with 4. It would be very useful to know how many heterostracans were examined from each of these individual HBHs. Perhaps I'm missing this somehow, but it would make most sense to go as a 5th row of data in the upper table of the tab of the spreadsheet called "Jawed Verts Occurrences". My concern here is that these three HBHs (of a total of 138, so roughly 2%) yield nearly two thirds of all reported bite marks. If these sites have very high numbers of heterostracans examined, then there may be nothing to worry about. However, if there are relatively low numbers of specimens from these HBHs--meaning exceptionally high frequencies of bite marks within these site relative to the overall frequency elsewhere--I would begin to wonder about what the pattern reported might actually mean, especially in terms of site-specific effects that are not really considered here (either in terms of the local biology of a given setting, past efforts in a particular set of localities that have prioritized collection of bitten material, etc.). If bite marks are concentrated within specific HBHs, this could also drive patterns of association with specific putative gnathostome predators. If this is indeed the case, I would encourage the authors to be frank about it, and provide some discussion of why this might be so and how it influences their inferences.

2) Related to the previous comment, is it possible to provide locality/place names for the HBHs? In the spreadsheet provided in the supplement, these are only given numerical names. However,

there is no straightforward way to tie these back to specific sites that would allow later researchers to amend/adjust/test the raw data shown here.

3) The analyses conducted here rely on measures of predation standardized for effort: the frequency of bitten specimens in any given time interval. While efforts are made to test the robustness of patterns to removal of individual stages, there is less consideration of the effects of single sites (see previous comment) or the relative uncertainty about the frequencies for individual stages. For example, are we really certain that the frequencies differ between some of these stages, given that they are subject to sampling error? For example, while it seems clear enough that the percentage (with no underlying information about sample sizes) of specimens showing tier 1 marks in the Pragian (1.7%) is lower than that of the Emsian (2.5%), which in turn is lower than the Frasnian (3.7%), are we confident those apparent differences are really meaningful when looking at underlying counts of 1 out of 60, 2 out of 79, and 3 out of 82? With such a low occurrence rate of bite marks and modest sample sizes, I suspect one would be hard pressed to demonstrate these and other frequencies are different from one another in a statistical sense. This can be--and perhaps should be--tested, and its implications for the results and conclusions considered.

4) The authors should be applauded for adding meaningful data to a debate that has largely relied on anecdote in the past. However, I encourage the authors to be more circumspect when it comes to what their data can test. While increasing evidence of bite marks over time is consistent with the hypothesis that gnathostome predation might have contributed to the extinction of ostracoderms, it still remains circumstantial. There is nothing wrong with this, as it indicates the predation hypothesis is still one worth pursuing. I think the study would benefit from some unvarnished reflection on this.

5) Perhaps this is getting into the weeds, but would it be more appropriate to consider the traces found here as examples of failed predation? Would casting the results in this light cause any reconsideration of the signal reported here?

Review form: Reviewer 2

Recommendation

Accept with minor revision (please list in comments)

Scientific importance: Is the manuscript an original and important contribution to its field?

Good

General interest: Is the paper of sufficient general interest?

Good

Quality of the paper: Is the overall quality of the paper suitable?

Good

Is the length of the paper justified?

Yes

Should the paper be seen by a specialist statistical reviewer?

No

Do you have any concerns about statistical analyses in this paper? If so, please specify them explicitly in your report.

No

It is a condition of publication that authors make their supporting data, code and materials available - either as supplementary material or hosted in an external repository. Please rate, if applicable, the supporting data on the following criteria.

Is it accessible?

Yes

Is it clear?

Yes

Is it adequate?

Yes

Do you have any ethical concerns with this paper?

No

Comments to the Author

The authors have addresses almost all of my concerns.

I only marked minor revisions because the authors did not address my note about a power analysis. I would still like to see a power analysis done to verify the statistical power of the tests performed. However, I leave it to the editors whether that is necessary for publication.

Decision letter (RSPB-2019-1596.R0)

21-Aug-2019

Dear Dr Sansom:

Your manuscript has now been peer reviewed and the reviews have been assessed by an Associate Editor. The reviewers' comments (not including confidential comments to the Editor) and the comments from the Associate Editor are included at the end of this email for your reference. As you will see, the reviewers and the Editors have raised some concerns with your manuscript and we would like to invite you to revise your manuscript to address them. Both reviewers were left unsettled by the robustness of the results even after revision and this is holding up publication; if not resolved it may lead to rejection. Hence it is vital to make further strides on this issue in revision, but this is deemed feasible.

To submit your revision please log into <http://mc.manuscriptcentral.com/prsb> and enter your Author Centre, where you will find your manuscript title listed under "Manuscripts with

Decisions." Under "Actions", click on "Create a Revision". Your manuscript number has been appended to denote a revision.

Research ethics:

Use of animals and field studies:

All supplementary materials accompanying an accepted article will be treated as in their final form. They will be published alongside the paper on the journal website and posted on the online

figshare repository. Files on figshare will be made available approximately one week before the accompanying article so that the supplementary material can be attributed a unique DOI. Please try to submit all supplementary material as a single file.

Please submit a copy of your revised paper within three weeks. If we do not hear from you within this time your manuscript will be rejected. If you are unable to meet this deadline please let us know as soon as possible, as we may be able to grant a short extension.

Best wishes,

Professor John Hutchinson, Editor
mailto:proceedingsb@royalsociety.org

Associate Editor Board Member

Comments to Author:

Randle and Sansom provide a comprehensive re-analysis of predation traces on heterostracans, a group of jawless fishes, and should be commended on their efforts. The authors find that the proportion of bite marks increased towards the end of the Frasnian, which they suggested was linked to the increasing prevalence of jawed fishes.

The two reviewers, although positive, still raised a number of significant issues that need to be addressed prior to publication. The concerns centre primarily around how robust are the patterns identified by the authors. Reviewer 1 asked for a power analysis to show effect size, whereas Reviewer 2 provided several suggestions for subsampling to ensure patterns are impervious to perturbations. Reviewer 2 also provided a number of other useful comments throughout the PDF, which is attached. I would urge the authors to carefully consider how they present the analyses, providing appropriate caveats and verbiage throughout the manuscript.

Reviewer(s)' Comments to Author:

Referee: 2

Comments to the Author(s).

The authors have addresses almost all of my concerns.

I only marked minor revisions because the authors did not address my note about a power analysis. I would still like to see a power analysis done to verify the statistical power of the tests performed. However, I leave it to the editors whether that is necessary for publication.

Referee: 1

Comments to the Author(s).

This contribution examines the frequency of inferred predation traces on heterostracan (armored jawless fish bones) over the latest Silurian and Devonian, with a goal of establishing whether such marks--and by extension, the intensity of predation pressure exerted by jawed fishes--increased over this interval. The authors finds an increase in bite mark frequency over time, and concludes that increased predation remains as a possible driver of extinction in 'ostracoderms'.

While I think that this study delivers some much-needed data bearing on a classic 'just-so' story in vertebrate paleontology, I think it would benefit from a bit more reflection the (i) limitations of the data at hand, (ii) the degree to which any result might constrain our understanding of 'ostracoderm' extinction, and (iii) slightly more complete supplementary material.

Specific comments are on a marked copy of the MS, and more general questions/comments are here.

1) More information should be provided to readers to determine if there is a site-specific pattern. In terms of overall counts of tier 1 marks within individual horizons, there are Givetian sites with 10 and 5 such traces, and one Lochkovian one with 4. It would be very useful to know how many heterostracans were examined from each of these individual HBHs. Perhaps I'm missing this somehow, but it would make most sense to go as a 5th row of data in the upper table of the tab of the spreadsheet called "Jawed Verts Occurrences". My concern here is that these three HBHs (of a total of 138, so roughly 2%) yield nearly two thirds of all reported bite marks. If these sites have very high numbers of heterostracans examined, then there may be nothing to worry about. However, if there are relatively low numbers of specimens from these HBHs--meaning exceptionally high frequencies of bite marks within these site relative to the overall frequency elsewhere--I would begin to wonder about what the pattern reported might actually mean, especially in terms of site-specific effects that are not really considered here (either in terms of the local biology of a given setting, past efforts in a particular set of localities that have prioritized collection of bitten material, etc.). If bite marks are concentrated within specific HBHs, this could also drive patterns of association with specific putative gnathostome predators. If this is indeed the case, I would encourage the authors to be frank about it, and provide some discussion of why this might be so and how it influences their inferences.

2) Related to the previous comment, is it possible to provide locality/place names for the HBHs? In the spreadsheet provided in the supplement, these are only given numerical names. However, there is no straightforward way to tie these back to specific sites that would allow later researchers to amend/adjust/test the raw data shown here.

3) The analyses conducted here rely on measures of predation standardized for effort: the frequency of bitten specimens in any given time interval. While efforts are made to test the robustness of patterns to removal of individual stages, there is less consideration of the effects of single sites (see previous comment) or the relative uncertainty about the frequencies for individual stages. For example, are we really certain that the frequencies differ between some of these stages, given that they are subject to sampling error? For example, while it seems clear enough that the percentage (with no underlying information about sample sizes) of specimens showing tier 1 marks in the Pragian (1.7%) is lower than that of the Emsian (2.5%), which in turn is lower than the Frasnian (3.7%), are we confident those apparent differences are really meaningful when looking at underlying counts of 1 out of 60, 2 out of 79, and 3 out of 82? With such a low occurrence rate of bite marks and modest sample sizes, I suspect one would be hard pressed to demonstrate these and other frequencies are different from one another in a statistical

sense. This can be--and perhaps should be--tested, and its implications for the results and conclusions considered.

4) The authors should be applauded for adding meaningful data to a debate that has largely relied on anecdote in the past. However, I encourage the authors to be more circumspect when it comes to what their data can test. While increasing evidence of bite marks over time is consistent with the hypothesis that gnathostome predation might have contributed to the extinction of ostracoderms, it still remains circumstantial. There is nothing wrong with this, as it indicates the predation hypothesis is still one worth pursuing. I think the study would benefit from some unvarnished reflection on this.

5) Perhaps this is getting into the weeds, but would it be more appropriate to consider the traces found here as examples of failed predation? Would casting the results in this light cause any reconsideration of the signal reported here?

Author's Response to Decision Letter for (RSPB-2019-1596.R0)

See Appendix B.

RSPB-2019-1596.R1 (Revision)

Review form: Reviewer 1

Recommendation

Accept with minor revision (please list in comments)

Scientific importance: Is the manuscript an original and important contribution to its field?

Good

General interest: Is the paper of sufficient general interest?

Good

Quality of the paper: Is the overall quality of the paper suitable?

Acceptable

Is the length of the paper justified?

Yes

Should the paper be seen by a specialist statistical reviewer?

No

Do you have any concerns about statistical analyses in this paper? If so, please specify them explicitly in your report.

No

It is a condition of publication that authors make their supporting data, code and materials available - either as supplementary material or hosted in an external repository. Please rate, if applicable, the supporting data on the following criteria.

Is it accessible?

Yes

Is it clear?

Yes

Is it adequate?

Yes

Do you have any ethical concerns with this paper?

No

Comments to the Author

I thank the authors for addressing previous comments. I have provided some comments on the attached .pdf.

Decision letter (RSPB-2019-1596.R1)

04-Nov-2019

Dear Dr Sansom:

Your manuscript has now been peer reviewed and the reviews have been assessed by an Associate Editor. The reviewers' comments (not including confidential comments to the Editor) and the comments from the Associate Editor are included at the end of this email for your reference. As you will see, the reviewers and the Editors have raised some concerns with your manuscript and we would like to invite you to revise your manuscript to address them. The reviewer and Associate Editor still feel the claims are overstated, but also that the paper has merit.

When submitting your revision please upload a file under "Response to Referees" in the "File Upload" section. This should document, point by point, how you have responded to the reviewers' and Editors' comments, and the adjustments you have made to the manuscript. We

require a copy of the manuscript with revisions made since the previous version marked as 'tracked changes' to be included in the 'response to referees' document.

Research ethics:

Use of animals and field studies:

If you wish to submit your data to Dryad (<http://datadryad.org/>) and have not already done so you can submit your data via this link [http://datadryad.org/submit?journalID=RSPB&manu=\(Document not available\)](http://datadryad.org/submit?journalID=RSPB&manu=(Document%20not%20available)), which will take you to your unique entry in the Dryad repository.

Online supplementary material will also carry the title and description provided during submission, so please ensure these are accurate and informative. Note that the Royal Society will

not edit or typeset supplementary material and it will be hosted as provided. Please ensure that the supplementary material includes the paper details (authors, title, journal name, article DOI). Your article DOI will be 10.1098/rspb.[paper ID in form xxxx.xxxx e.g. 10.1098/rspb.2016.0049].

Please submit a copy of your revised paper within three weeks. If we do not hear from you within this time your manuscript will be rejected. If you are unable to meet this deadline please let us know as soon as possible, as we may be able to grant a short extension.

Best wishes,
Professor John Hutchinson
Editor, Proceedings B
mailto:proceedingsb@royalsociety.org

Associate Editor
Board Member: 1
Comments to Author:

The authors have revised their manuscript sufficiently well to address the reviewer's concerns. The language they use is more conservative now regarding their conclusions on predation-driven extinction, which I think is appropriate given the nature of their data (only 41 predated specimens). While I remain unconvinced of their hypothesis, I think this is a useful contribution to inspire debate and further research on the topic. Aside from the Reviewer's remaining comments and suggestions for changes (see attached PDF), I have a few more listed here:

The authors need to add verbiage in the Discussion regarding the validity of the other hypotheses, which they did not test. For example, environmental changes may well have contributed to the extinction of jawless fishes. This hypothesis could even be tested in a similar fashion to the biotic hypothesis tested here. I am not suggesting the authors do this, but merely that it could be done, and their analyses (as they stand) do not put a nail in the coffin of this explanation. Moreover, as the authors know, correlation does not equal causation, and this should be reiterated in the Discussion. Even if predation of jawless fishes by jawed fishes increased through time, it doesn't necessarily mean it caused the demise of jawless fishes. Verbiage of this nature could potentially go after line 255.

Lines 98-99: the methodology should be clarified here

Line 126: please remove the comma at end of this sentence

Line 145: need a 'with' here?

The Results section could benefit from some subheadings; the section from 158-183 could benefit from paragraph breaks.

Line 226: perhaps better worded as 'show a significant increase through time'

Line 244: phrasing here needs to be revised

Reviewer(s)' Comments to Author:

Referee: 1

Comments to the Author(s)

I thank the authors for addressing previous comments. I have provided some comments on the attached .pdf.

Author's Response to Decision Letter for (RSPB-2019-1596.R1)

See Appendix C.

Decision letter (RSPB-2019-1596.R2)

19-Nov-2019

Dear Dr Sansom

I am pleased to inform you that your manuscript entitled "Bite marks and predation of fossil jawless fish during the rise of jawed vertebrates" has been accepted for publication in Proceedings B. Congratulations!!

Open Access

Your article has been estimated as being 9 pages long. Our Production Office will be able to confirm the exact length at proof stage.

Paper charges

Sincerely,

Professor John Hutchinson
Editor, Proceedings B
mailto: proceedingsb@royalsociety.org

Appendix A

The University of Manchester
School of Earth and Environmental Sciences
University of Manchester
Oxford Road, Manchester
M13 9PT

robert.sansom@manchester.ac.uk

2.07.2019

Dear Prof. Hutchinson,

Thank you for your response regarding RSPB-2019-0585 entitled "Bite marks and predation of fossil jawless fish during the rise of jawed vertebrates". The editorial board member and three reviewers were very positive about the submission and highlighted some useful revisions to strengthen the contribution. We thank all four for their input and have revised the manuscript in light of these suggestions. In summary these revisions primarily relate to:

- Statistical analyses. The diversity results were standardized for sampling and correlations were found to be robust to additional jackknife and first differences tests.
- Analyses at lower taxonomic levels. In addition to sub-clades of jawed vertebrates, we have also considered them at genus level in terms of frequency of co-occurrence with bite marks and correlation with mandible length.
- Clarification over our approach to data samples. We have provided more detail on our data collection and demonstrate that the analyses of the complete bite mark dataset and the unambiguous bite mark dataset (tier 1) show the same results.

These revisions have enabled us to strengthen the manuscript and have given us more confidence in findings i.e. that predation of jawless vertebrates increasing through time, and that this predation is related to the distribution and diversity of jawed vertebrates. It provides the first direct evidence to test hypotheses relating to a pivotal moment in evolutionary history, the replacement and extinction of armoured jawless vertebrates by jawed vertebrates. As such, we believe that this study will be of broad interest to the palaeontological and evolutionary research communities as well as broader scientific and public audiences.

Detailed response to the comments can be found below. The edited sections of the manuscript are highlighted in an uploaded revised version.

Your Sincerely

Robert Sansom and Emma Randle

Associate Editor

Board Member: 1

Comments to Author:

Randle and Sansom provide a comprehensive and impressive survey of predation traces on heterostracans, a group of jawless fishes, from the Silurian to Devonian. The authors found that the proportion of bite marks increased towards the end of the Frasnian, which they suggested was linked to the increasing prevalence of jawed fishes. Therefore, the decline of jawless fishes was potentially driven by predation from their jawed cousins. The paper is well written, the presentation is straightforward, and I commend the authors on what was clearly a large amount of work. All three reviewers, however, raised a number of issues that need to be addressed prior to publication. I also detail some additional comments below, which may further improve this already-excellent contribution.

The statistical tests were mentioned as needing improvement by all three reviewers, and I concur. For example, the authors need to (at least) test whether their time series correlations are stationary (e.g., using the Augmented Dickey-Fuller Test). If their data are not stationary, first differencing would be appropriate, as mentioned by Reviewer 3. Correlation coefficients (in addition to p-values) need to be provided in the text so that readers can gauge the strength of reported relationships. Finally, Reviewer 2 provided good suggestions to examine the robustness of patterns, such as jackknife and power analyses.

Response: We thank the editorial board and reviewer comments regarding the statistical analyses. In the revised version of the manuscript we have included a wider range of data analyses which have strengthened the contribution and its conclusions. We have accounted for the possibility of auto-correlation of our time series by including a first differences analysis. The correlations between bitemark occurrence and jawed vertebrate diversity were also found to be robust and significant following jackknifing analyses and first differences analysis (and correlation values are reported in the text).

The authors correlated jawed vertebrate diversity against heterostracan bite prevalence through time, but I wondered how this diversity was calculated? That is, did the authors use raw diversity estimates, or was diversity subsampled/standardized? If only the former, I would suggest examining patterns using the latter, since this would help isolate whether diversity is simply a correlate of available sedimentary packages (for example).

Response: We thank the editor for this suggestion and agree with premise. Rather than raw diversity counts, the revised version of the manuscript uses standardised diversity (using the number of heterostracan bearing formations for sampling). The overarching patterns remain the same.

The authors discussed tier 1 versus tier 2 bite marks, but it was unclear from reading the text what constituted a tier 2 bite mark, and how tier 2 bites were distinguished from tier 1 bite marks. This should be further explicated. Moreover, Reviewer #3 raised the question of why all jawed vertebrate guilds are included in the diversity analyses, when some of these groups (e.g., Acanthodes) cannot eat jawless fishes.

Response: In the revised version we have clarified our distinction between tier 1 and tier 2 and have focused the discussion of tier 1 (those identified with a high degree of confidence). The revised text is below. We have avoided identifying 'guilds' of jawed vertebrate genera a

priori because classifications are relatively subjective, not aligning with taxonomy, and could be disputed. We have, however, followed a suggestion from reviewer 1 and used mandible length of jawed vertebrates genera as a quantitative proxy for predation potential (being related to both of both body size and jaw gape). In the revised version: “Bite occurrences were collected from direct observations of specimens and from the literature and placed in two tiers relating to confidence of identification: tier 1 comprises novel identifications of traces with constrained morphologies that meet multiple bite mark criteria (Fig. S1) as well as traces previously described in the literature [4, 13-18]; traces that met only some of the specific bite mark criteria or occurred in otherwise fragmentary specimens were classified as tier 2 to reflect their more tentative assignment as bite marks. As such, tier 1 traces are those unambiguously interpreted as bite marks following application of their morphology and preservation to the precise identification criteria (1-6 above) whilst tier 2 traces were included in a broader dataset retained given their potential evidence of predation. We take two approaches to data analysis: the complete bite mark dataset, or just tier 1 dataset.”

*Fig S3 doesn't seem to be referenced in the main text, and I wasn't clear what this figure depicted. Did authors examine predation intensity in jawed vertebrates within HBFs, in addition to predation intensity in jawless fishes? If so, the total number of specimens examined should be included here, as was done in Fig 2. Moreover, what do patterns look like through time? Does the prevalence of bite marks for jawed vertebrates also increase? If yes, would this affect the authors' conclusions? I was concerned that one of the strongest relationships between predation frequency and jawed vertebrates was for the placoderm *Asterolepis*, which the authors indicated would have been unlikely to predate jawless fishes (lines 63-68). I would suggest that all taxa that are not candidate predators be removed from all analyses and patterns re-analyzed.*

Response: In the revised version we have changed our approach measuring the frequency of co-occurrence of bite mark traces and jawed vertebrates by accounting for sampling bias. We have also explicitly discussed the analysis and findings in the text. *Asterolepis* was initially identified because it simply is the most common jawed vertebrate in our dataset. In the standardized dataset, more likely predators such as *Panderichthys* are associated with bite marks given their frequency of co-occurrence. This is represented in the new Fig. 4.

The analyses suggested by Reviewer 1, which focus on examination of gape size and body size in relation to predation intensity, would broaden the reach of this contribution even further.

Response: We have now followed this suggestion (see below).

Line 15: this is the first time heterostracans are mentioned, so please define - **Amended**

Lines 24-26: consider toning down the confidence level for the last sentence of the abstract. - **Amended**

Line 46: remove 'a' prior to benthic mud grubbers - **Amended**

Line 62: remove 's' on vertebrates - **Amended**

Line 166: need a space between parentheses - **Amended**

Line 184: extra space between 'skeleton' and 'was' - Amended

Line 188: comma before 'but eurypterid...' - Amended

Fig 2, part b: a blue dot appears at -430 (x-axis) above the figure legend. Is this an occurrence? - Amended

Fig 3, part c: I am assuming the y-axis values are proportions? If so, this should be noted. - Amended

Reviewer(s)' Comments to Author:

Referee: 1

Comments to the Author(s)

The decline of armored jawless fishes and the associated rise of jawed fishes is one of the most conspicuous features of the early vertebrate fossil record, and set the stage for the subsequent 400 million years of evolution in backboned animal evolution. The causes of this stark change in faunas during the Devonian has been much debated, largely from the perspective of diversity metrics (i.e. taxonomic richness and morphological disparity). At best, such approaches offer circumstantial evidence for evaluating a range of competing hypotheses. One popular explanation for the decline of jawless fishes in the face of gnathostome diversification invokes direct interaction between these groups, in the form of increased predation pressure by jawed vertebrates. Past evidence for this has been, to put it generously, anecdotal and diffuse.

The authors provide a comprehensive survey of a particularly diverse group of jawless vertebrates (heterostracans) over their Silurian-Devonian record. They report direct evidence of (failed) predation in the form of bite marks on the armor plates of these jawless fishes, probably (but not certainly) left by jawed fishes. The authors clearly demonstrate two important relationships. First, that bite marks increase over the history of heterostracans, and second, that the occurrence of bite marks correlates with the co-occurrence of certain large groups of jawed fish predators. While this study does not provide a definitive answer to the extinction of armored jawless fishes (and the authors acknowledge this fully), it nevertheless provides an important line of evidence that has previously received shockingly little attention. I imagine that the plots presented here are the kinds of things that are likely to appear in lectures in vertebrate paleontology classes, and it would not be a stretch to imagine them finding a place in textbooks.

I have made a few suggestions in the text, but these largely concern wording and typographical errors. The most substantive changes I recommend in my in-text suggestions are:

(1) The use of non-parametric rank-order tests for establishing patterns of covariation (there is no reason to assume that the correlations here are linear). I am reasonably sure that the results presented here will be robust to this alternative approach.

Response: We thank the reviewer for this suggestion. In the revised version, we have used Spearman's rank correlations (a more appropriate non-parametric test) rather than Pearson's. The qualitative outcomes remain largely the same.

(2) *The use of taxonomically and stratigraphically appropriate group icons in Figure 3. The sarcopt icon is in fact an actinopt (Cheirolepis), while the use of a hammerhead shark and a paddlefish for chondrichthyans and actinopts is about as phylogenetically and stratigraphically incongruous as alternatives like a torpedo ray and seahorse.*

Response: We thank the reviewer for this suggestion. In the revised version, we have used different icons.

There are a few additional things I would ask the authors to consider, and which would only really require the addition of a line or two to the text. It would be very easy to add something more substantive, since these data are available 'off the shelf'. I think this would add considerable value to this paper:

(1) *At least an consideration of body size. One of the most striking things (to my mind, anyhow) is the very large size of some of the youngest heterostracans. These data are readily available, in the supplement of Sallan & Galimberti (2015, Science). There are obviously many reasons why body size might increase within a group over time, but it seems like this is something worth mentioning as a possible (but untested) response to predation.*

Response: This is an interesting suggestion. Increases in heterostracan body size over time, or relative to a phylogenetic framework, would certainly be of interest, but difficult to justify in this contribution given the lack of a direct link between bitemarks and prey body size given other possible conflating factors. We have instead used mandible size (see below).

(2) Related to the above, it would be interesting to consider whether gape size increased in gnathostomes over this same interval. The data are available in the supplement to Anderson et al. (2011, Nature).

Response: We thank the reviewer for this suggestion. In the revised version of the manuscript we have explicitly considered the gape size (using mandible length) of jawed vertebrate genera by testing its correlation with frequency of co-occurrence of bitemark type traces through time. We find a positive, but non-significant correlation between gape size (mandible size) and frequency of co-occurrence with bite marks (Fig. 4):

“Of the 198 jawed vertebrate genera co-occurring with heterostracans, mandible length data were available for 53 [2]. There was a positive correlation between jawed vertebrate mandible length and frequency of co-occurrence with tier 1 bite mark specimens (Spearman’s rho $\rho=0.23$) but this was not significant ($p=0.098$) (Fig. 4)”.

Referee: 2

Comments to the Author(s)

The authors present bite mark data on jawless vertebrates from the Silurian and Devonian periods to address the question of predation as a potential driver for the reduction in jawless vertebrate diversity towards the end Devonian. They examine the number of specimens showing “bite marks” relative to the total number of specimens examined across time bins and see an increase in bite marks towards the end Frasnian. They use these results to argue that increased predation may be a cause of jawless vertebrate diversity reduction.

I appreciate the attempt to use an oft-overlooked form of trace fossil data (predation marks) to address broader questions in paleoecology. However, I find the data presented too sparse to make the conclusions that the authors are trying to make. They looked at over 2900 specimens and found, at most, 41 bite mark cases. Meaning they only found predation evidence on less than 2% of the specimens examined. Furthermore, some of these are classified as “tier 2” meaning they don’t actually fit the criteria laid out in the methods. The authors don’t give enough detail about what constitutes a “tier 2” bite mark, making it unclear if these really are evidence of predation. Two of the tier-2 instances are noted to not come from localities that have associated gnathostome fossils, meaning there is no evidence of jawed predators. At the very least, these two specimens should be removed from the analysis. I would suggest only including tier-1 bite marks (note, the authors do show tier 1 only results as well), but this reduces the amount of actual data to less than 1% of the specimens examined.

Response: The reviewer is correct that occurrence of bitemarks is rare. Indeed preservation of direct evidence predation would be expected to be rare. The important conclusions we draw relate to the **relative** occurrence of those bitemarks through time and in relation to jawed vertebrate occurrences and diversity. Furthermore, this relationship is robust (see comments above and below). We have clarified our description and analysis of tier 1 and tier 2 traces and focused the discussion of tier 1 (see revised text above in response to editor comments). Fundamentally, tier 1 are unambiguous bite marks meeting multiple criteria whilst tier 2 meet some criteria and are more tentatively interpreted. Tier 2 data have been retained so that we are open about the data - the results remain the same for the complete dataset or for the focused dataset. Furthermore, we don’t think it would be appropriate to remove certain bite-mark type traces a priori because they do not co-occur with jawed vertebrates known to date; this could introduce circularity when testing the hypothesis that jawed vertebrates are responsible.

Amount of data aside, the pattern presented in figure 2, meant to show increasing bite marks through time, is not convincing. The trend is partly driven by the first three bins showing essentially no bite marks across the largest number of specimens examined. The Wenlock has almost 1000 specimens checked by itself (a third of the total data set) and only a single, tier-2 bite mark. These bins are certainly going to exert an inordinate amount of influence on the regression, but it is unclear whether there even were gnathostome predators around. The one bite mark found in the Wenlock is tier-2, and I assume one of the two that isn’t associated with gnathostomes. A lack of predators in these early bins is likely a real signal, but if the goal of this study is to assess whether predation itself increased, only localities that include both jawless and jawed vertebrates should be used. This means removing all of the jawless specimens from their count of 2900 that appear without co-occurring jawed forms.

Response: The occurrence of bite mark type traces increases through time. This is true of the complete dataset or just tier 1 dataset. For the revised manuscript we have applied jackknife tests to these relationships and they were found to be robust. To test the hypothesis that jawed vertebrates could be the predators responsible, we tested against jawed vertebrate occurrence and diversity. It is therefore necessary and appropriate to include time bins for which jawed vertebrates do not co-occur with heterostracans. Jawed vertebrates have been recorded from those early time bins (e.g. Wenlock), but were found not to co-occur with heterostracans in our dataset. From the revised text: “This relationship is significant for the combined dataset of all bite mark types (Spearman’s rho $\rho=0.86$, $p=0.003$, $n=9$) and for tier 1 bite marks (Spearman’s rho $\rho=0.76$, $p=0.018$, $n=9$). Both relationships are

robust to first-order jackknife tests (removal of any individual time bin gives correlation $p < 0.05$ for complete dataset, and $p < 0.05$ for 7 of 9 time bins for the tier 1 only dataset).”

Figure 2 also shows that the signal in the later Devonian seems to be strongly influenced by one time bin, the Givetian. It is unclear how much isolated localities may be effecting these numbers, but that is another aspect of the data that must be examined.

There are a series of steps the authors must take to alleviate these issues:

- 1) Remove tier 2 data from the analysis, or at least the two examples that are not associated with any gnathostome fossils. In general, if the interest is in the trend of predation, no jawless specimens from localities that lack gnathostomes should be used
- 2) A statistical power analysis is vital to convince the reader that there is enough data to actually support the proposed trend.
- 3) The authors should perform a jackknifing procedure to assess the influence of exceptional fossil assemblages that may be biasing the results; particularly in the Givetian, which appears to have more bite mark evidence than any other bin including the later Frasnian.

These steps are vital to determining whether the data is sufficient to actually support the proposed trend.

Response: We thank the reviewer for these suggestions. In the revised version of the manuscript, we have clarified our identification of tier 1 and tier 2 bite mark traces (see response above). We have focused the discussion on tier 1 trace fossils (i.e. unambiguous bite marks) but we have retained the complete dataset analysis (i.e. tier 1 and tier 2 combined) to be open about our data and to see if there is any difference in the signal conveyed by these two classes of data - there is not. To address the statistical power and robustness of the result, we have applied first differences analyses to eliminate any possible time-series autocorrelation (see above), we have applied non-parametric tests instead of parametric tests (Spearman's rho instead of Pearson's correlation), we have standardised the jawed vertebrate occurrences for sampling (see above), and we have applied first order jackknife analyses with elimination of each individual time bin for all of the correlation tests. The qualitative results remain largely the same for both classes of data and are robust to these extra battery of tests. From the revised text: “The combined bite mark prevalence data (tier 1 and tier 2) shows the same pattern of significant correlations as for tier 1 prevalence alone. Applying first order jackknifing (removal of each time bin) found these correlations to be robust for the combined prevalence (complete dataset). jackknifing tier 1 bite mark prevalence found only the correlation with sarcopterygian diversity to be robust, but the number of specimens is much smaller. “

A couple other issues:

Discussion, Pg. 6, Ln. 167-172: It is noted that the most likely predators of jawless fish at the time would be acanthodians, sarcopterygians and actinopterygians. However, the data shows that only the placoderms and sarcopterygians were significantly associated with bite marks. This further undercuts the conclusions as the placoderms that seemingly drive much of the co-occurrence data are considered NOT to be viable predators.

Response: In the revised version we have expanded the analyses and discussion of possible trace makers. Relating to the specific point of the reviewer, both acanthodians and placoderms show large diversity of jaw sizes morphologies, and thus vary in their predator potential within those groups. We have taken therefore taken two approaches - grouping the jawed vertebrate genera taxonomically, or considering them as individual genera in terms of

their frequency of co-occurrence with bitemarks and their mandible size. From the revised text: “The individual jawed vertebrate taxa most frequently co-occurring with heterostracan specimens with bite marks include *Panderichthys*, *Livosteus*, and *Grossipterus*. All are large, with well developed jaws and teeth making them potential predators of jawless vertebrates.”

Figure 3: The Y-axis for part C are labeled as “number of predation traces”. However, these axis have values of less than 1. Presumably these are standardized values? That should be made explicit in the label.

Response: We thank the reviewer for this comment. The figure has been revised.

Referee: 3

Comments to the Author(s)

Randle and Sansom have produced a very interesting study of the incidence of predation on jawless fishes by their jawed cousins. Direct evidence for this kind of predation has not yet been fully documented outside scattered observations, or quantified, preventing further discussion and explicit tests of the role of jawed vertebrates in jawless fish decline and extinction. The authors have done a thorough job of combing museum collections and identifying predation traces in one group of armored fishes, the heterostracans, according the standards of the field, and have made a solid attempt to connect the incidence of predation damage with the presence of different jawed predators. This is an excellent and much needed effort, but I have some concerns about whether the statistical tests and comparisons used are the most appropriate or necessary. Thus, I recommend something like moderate revisions to the methods, even though that's not an option on the pull-down menu. However, I don't anticipate this will take long or require a large effort on the author's part, and I eagerly look forward to seeing this important study published!

Methods and Use of Statistics:

*I am not sure all the statistical tests are necessary or appropriate given the clarity of the general pattern. That said, some of the variables need to be better standardized and aligned with the hypotheses. For example, It seems like a bit of an “apples to oranges” comparison to use absolute number of predation traces (which may be made by a specialized single predator) and genus diversity (which includes all guilds, even those which cannot eat jawless fishes like *Acanthodes* or *Bothriolepis*) in the tests in Fig. 3. That is, it's not clear what the relationship should be, or what would drive a significant correlation. It might be better to test something like the association between numbers with bite marks/percentage with bite marks and number of jawed fish specimens from the same site/percentage of taxa which are jawed. That way the variables are in the same general class.*

Response: We thank the reviewer for this suggestion and agree with the premise. In the revised version of the manuscript, the number jawed vertebrate genera recovered from heterostracan bearing formations in each time bin has been standardised for sampling. As such, the data are more comparable for each. The qualitative results are unchanged. We have also taken two approaches by grouping the jawed vertebrates taxonomically, and in the revised version, considering them at the level of individual genera in terms of frequency of co-occurrence with bite marks as well as jawed vertebrate mandible length (see response comments above).

You could also compare number of distinct types of predation traces (punctures or damage) with genus diversity, because then it would be number of kinds of attackers vs. number of kinds of attacks. But in that case, the diversity of jawed fishes should be limited to those with biting predation and/or the kinds of teeth which can produce observed damage (e.g. taxa with coronoid fangs rather than durophages).

Response: We thank the reviewer for this suggestion. Whilst this is an interesting idea, it would not be practicable given the sample size of the bite-mark type traces.

In all correlation tests of time series, first differencing should be used rather than the full value to prevent autocorrelation (see Sallan et al. 2011). Then you are explicitly testing the correlation between the direction of change, and reduce any effects of large increases in jawed fish diversity and abundance. Is the positive correlation with sarcopts and placoderms driven by greater sample size for those groups in an interval of increased gnathostome dominance?

Response: We thank the reviewer for this suggestion. In the revised version we have applied first differences tests to eliminate possible autocorrelation of the time series data. The results are robust. From the revised text: “To account for possible auto-correlation of time series data, first-differences tests were applied and the same correlations were significant ($\rho=0.80$, $\rho=0.012$, $\rho=0.92$, $\rho=0.001$, $\rho=0.90$, $\rho=0.002$ for all jawed vertebrates, placoderms and sarcopterygians diversity with heterostracans with bite mark prevalence respectively)”.

It is not clear that the Chi Square tests of predation trace occurrence and gnathostome presence add anything given that all with tier 1 predation traces have jawed fishes, that involving tier 2 removes significance, and the small sample sizes relative to the number of sites examines. It's probably enough to just note the general pattern.

*An alternative to the regression and chi-square tests would be to test if there are any predatory genera/families/clades/predation categories more closely associated with the 11/16 sites with bite marks vs. the 127/122 without. Some likely taxa are noted in the discussion (lines 165-166) but this could be tested explicitly, perhaps revealing other associations. Appropriate tests would include ANOSIM and SIMPER (see the book *Paleontological Data Analysis* by Hammer and Harper). Your supplementary data sheet “Jawed Vertebrate HBH” is almost in the right format for doing this at the genus or group level. It might just require some rearrangement of the rows into groups with extinction traces and without overall or by stage.*

Response: We thank the reviewer for this suggestion. In the revised version we have investigated the individual genera in terms of their frequency of co-occurrence in HBH with bite marks. We have also applied community analysis (a permutational MANOVA akin to those suggested by the reviewer) but found no significant difference between the jawed vertebrate fauna in HBH with bite mark specimens and in HBH without bite mark specimens.

Discussion:

Is it possible Tier 2 traces were produced by something else, such as arthropods or mollusks given the absence of jawed fishes at two of the sites? This possibility is raised briefly in the discussion but not made explicit. How do these traces compare to those assumed to be eurypterid as noted in lines 188-189?

The discussion mentions that most of the traces are likely sublethal. It would be good to know explicitly about how many traces were fatal vs. non-fatal, given evidence for regrowth.

Response: 10 of the 41 specimens exhibit re-growth (criterion 4, presented in the table). In the revised version we have added this to the text.

Another good paper dealing with change in gnathostome traces, and identification of fatalities, through time is Salamon et al. Paleobiology 2013

Response: We thank the reviewer for pointing out this relevant reference. It has been added to the revised text.

Appendix B

The University of Manchester
School of Earth and Environmental Sciences
University of Manchester
Oxford Road, Manchester
M13 9PT

robert.sansom@manchester.ac.uk

2.10.2019

Dear Prof. Hutchinson,

Thank you for your response regarding RSPB-2019-1596 entitled "Bite marks and predation of fossil jawless fish during the rise of jawed vertebrates". The editorial board member and reviewers were positive about this revised version and suggested some further revisions before publication was recommended. These principally related to the statistical analyses which we have now included and discussed.

We thank the reviewers and editors as these revisions have enabled us to strengthen the manuscript. It provides the first direct evidence to test hypotheses of predation relating to a pivotal moment in evolutionary history i.e. the replacement and extinction of armoured jawless vertebrates by jawed vertebrates. As such, we believe that this study will be of broad interest to the palaeontological and evolutionary research communities as well as broader scientific and public audiences.

Detailed response to the comments can be found below. The edited sections of the manuscript are highlighted in an uploaded revised version.

Your Sincerely

Robert Sansom and Emma Randle

Associate Editor Board Member Comments to Author:

Randle and Sansom provide a comprehensive re-analysis of predation traces on heterostracans, a group of jawless fishes, and should be commended on their efforts. The authors find that the proportion of bite marks increased towards the end of the Frasnian, which they suggested was linked to the increasing prevalence of jawed fishes. The two reviewers, although positive, still raised a number of significant issues that need to be addressed prior to publication. The concerns centre primarily around how robust are the patterns identified by the authors. Reviewer 1 asked for a power analysis to show effect size, whereas Reviewer 2 provided several suggestions for subsampling to ensure patterns are impervious to perturbations. Reviewer 2 also provided a number of other useful comments throughout the PDF, which is attached. I would urge the authors to carefully consider how they present the analyses, providing appropriate caveats and verbiage throughout the manuscript.

Response: We thank the editorial board member for this feedback. In the revised version we have included additional statistical analyses to assess the robustness of the results (which in the most part they are). We have revised the presentation of the results, including caveats. Details responses can be found below.

Referee: 2 Comments to the Author(s).

The authors have addresses almost all of my concerns. I only marked minor revisions because the authors did not address my note about a power analysis. I would still like to see a power analysis done to verify the statistical power of the tests performed. However, I leave it to the editors whether that is necessary for publication.

Response: We have included a power analysis in the revised version of the manuscript. "Power analysis (SI fig. 3) found that Spearman's rank correlation coefficient analyses with sample size of $n=9$ have high power for high correlation results only ($\rho>0.70$); the correlation analyses found to be robust to first-differences analyses and jackknifing fall within this range."

Referee: 1 Comments to the Author(s).

This contribution examines the frequency of inferred predation traces on heterostracan (armored jawless fish bones) over the latest Silurian and Devonian, with a goal of establishing whether such marks--and by extension, the intensity of predation pressure exerted by jawed fishes--increased over this interval. The authors finds an increase in bite mark frequency over time, and concludes that increased predation remains as a possible driver of extinction in 'ostracoderms'. While I think that this study delivers some much-needed data bearing on a classic 'just-so' story in vertebrate paleontology, I think it would benefit from a bit more reflection the (i) limitations of the data at hand, (ii) the degree to which any result might constrain our understanding of 'ostracoderm' extinction, and (iii) slightly more complete supplementary material. Specific comments are on a marked copy of the MS, and more general questions/comments are here.

1) More information should be provided to readers to determine if there is a site-specific pattern[...]

Response: We thank the reviewer for this suggestion. We agree that a site specific analysis using the number of heterostracan specimens from each heterostracan-bearing formation (HBH) would be of interest, but unfortunately it is not practicable. We were able to collate the specific site of origin for the 41 heterostracan specimens yielding bite mark type traces, but it would not be possible to do so for the other 2804 heterostracan specimens; not only are they too numerous, but the granularity of location data would not be available for many of those specimens. In the revised version we have undertaken a different kind of site specific analysis. We have randomly resampled the HBH for the co-occurrence chi-square tests and found the co-occurrence of placoderms and bite-mark type traces to be robust to data sampling.

2) Related to the previous comment, is it possible to provide locality/place names for the HBHs? [...]

Response: The revised version of the supplementary data now includes locality names.

3) *The analyses conducted here rely on measures of predation standardized for effort: the frequency of bitten specimens in any given time interval. While efforts are made to test the robustness of patterns to removal of individual stages, there is less consideration of the effects of single sites (see previous comment) or the relative uncertainty about the frequencies for individual stages. [...] This can be--and perhaps should be--tested, and its implications for the results and conclusions considered.*

Response: We thank the reviewer for this suggestion. As we stated in the response to point 1 above, resampling the data by site for prevalence analyses is not possible because the data for the number of specimens in each site is unavailable. Nevertheless, in the revised version we have included a site-specific analysis for the co-occurrence tests, acknowledge of and a new section discussing the rarity of the bite marks and the robustness of the statistical analyses (reproduced below).

“Bite marks were generally rare: 41 of 2846 heterostracan specimens sampled exhibited bite mark type trace (1.4% prevalence), and those bite marks occurred in 14 of 137 heterostracan bearing formations, across 7 of the 9 time bins. Rather than the absolute amount of bite marks, the significant patterns observed relate to their relative prevalence through time, their prevalence relative to jawed vertebrate diversity, and their co-occurrence with jawed vertebrates in horizons. Application of first-order jackknife tests shows that correlation results are generally robust for the increasing prevalence of bite mark type traces through time (removal of any individual time bin gives correlation of $p < 0.05$ for the complete dataset, and 7 of 9 time bins for the tier 1 only dataset), and for the correlations of prevalence of bite mark type traces with jawed vertebrate diversity (removal of any time bin gives correlations of $p < 0.05$ for all jawed vertebrates, sarcopterygians and placoderms for the complete dataset, but only sarcopterygians for the tier 1 bite marks). The first-difference analyses were also subjected to first-order jackknife tests and the originally significant relationships were largely found to be robust (removal of 8 of 9 time bins recovers significant correlations except the correlation between sarcopterygian diversity and tier 1 bite mark type traces). Power analysis (SI fig. XX) found that Spearman’s rank correlation coefficient analyses with sample size of $n=9$ have high power for high correlation results only ($\rho > 0.70$); the correlation analyses found to be robust to first-differences analyses and jackknifing fall within this range. The tests for co-occurrence of bite marks and jawed vertebrates in HBH were subjected to rarefaction analyses (resampling HBH with a probability of $\frac{2}{3}$, 100 times). The significant co-occurrence of bite-marks and placoderms in HBH was found to be extremely robust (97 and 100 of the 100 resampling iterations recovered chi-square $p < 0.05$ for the complete dataset and tier 1 only dataset respectively), but the other relationships were less robust (23 and 54 of the resampling iterations recovered significant co-occurrence of all jawed vertebrates and sarcopterygians respectively for the tier 1 dataset).”

4) *The authors should be applauded for adding meaningful data to a debate that has largely relied on anecdote in the past. However, I encourage the authors to be more circumspect when it comes to what their data can test. [...]*

Response: In the revised version we have updated the discussion and abstract to take a more circumspect tone (see below). In the original and revised version we considered predation alongside other possible causes (see further below).

Abstract: “Here we provide direct evidence of escalating predation from jawed vertebrates as a potential contributing factor to the demise and extinction of ostracoderms.”

Discussion: “The low absolute numbers of bite marks in this study (41 of 2846 specimens sampled) make it hard to draw definitive conclusions, but patterns of increasing prevalence through time and correlations with diversity of jawed vertebrate clades are robust to jackknifing and conservative first-differences tests. Furthermore, both tiers of trace fossil data (all bite marks, or just tier 1, unambiguous bite marks) show the same overarching results, and their co-occurrence with placoderms is robust to data resampling. The combined

analyses therefore suggest a possible role of jawed vertebrates in the predation of heterostracans.”

“Previous interpretations have suggested that ostracoderms suffered from restrictive environmental preferences [1,9] and limited dispersal capability [27] in this time of change. The data presented here add increasing pressure from predation by jawed vertebrates as a possible ~~cause of~~ factor in their decline and extinction.”

5) Perhaps this is getting into the weeds, but would it be more appropriate to consider the traces found here as examples of failed predation?

Response: In the manuscript we acknowledge that specimens destroyed or consumed through predation will go unrecorded, but the prevalence of bite-marks (including sub-lethal repair and thus ‘failure’) is direct evidence of predation.

Minor revisions from Reviewer 1 annotated PDF

L10: amended

L16: amended

L21: amended

L28: citation added

L33: amended

L32-33: How strong is the impact of the Frasnian/Famennian event on jawed fishes? My impression was that it was not substantial (particularly in comparison to the end-Devonian), although the use of the term ‘suffer’ implies otherwise.

Response: The Frasnian-Famennian boundary (also referred to as the Kellwasser Event) is not as drastic as the End Devonian Mass Extinction (often referred to as the Hangenberg Event), but the Kellwasser event saw the extinction of many vertebrate groups including jawless vertebrate clades, marine placoderms and many conodont clades. It is believed that although there is no significant difference in diversity over the Frasnian Famennian - there was a great degree of faunal and ecological turnover, associated with the creation of the supercontinent Pangea and increased cosmopolitanism of fishes (Friedman & Sallan 2012).

L48: amended

L50: amended

L51: But is this the hypothesis that is actually tested? Maybe outline the specific hypotheses that are tested as a list, and how these might support the idea that predation played an important role in extinction.

Response: Specific hypotheses added.

L58: amended

L78: amended

L78&81: amended

L89: amended

L90: amended

L111: amended

L113: amended

L121: see response to comment 3 above

L128: amended

L132: amended

L150: This also seems to carry with it some biogeographic or environmental signal: aren't most of these taxa from the Baltic? Is this a consequence of a set of geographically proximal sites driving this pattern?

Response: Additional analysis added by region: “These four taxa are all present in the baltic region only, but patterns appear unrelated to geography: other proximal baltic sites yield the inverse, and other regions yield high numbers of bite marks (6 from the Welsh borderlands). Furthermore, there is no significant relationship between region and presence of bite marks type traces (ANOVA $F=0.48$, 0.61 $p>0.9$ for $n=138$ HBH in 31 regions for tier 1 only and complete dataset respectively).”

L159: How were these differences between faunas quantified? Not clear from the description.

Response: Described in methods.

L169: What is the meaning of the light grey test statistics and p values?

Response: Figure legend specifies light grey as total dataset, black as tier 1 only dataset.

L189: amended

L198: See geographic response above.

L201: amended

L217: amended

L223 How does high disparity indicate turnover?

Response: The studies referred to discuss turnover.

Appendix C

MANCHESTER
1824

The University of Manchester
School of Earth and Environmental Sciences
University of Manchester
Oxford Road, Manchester
M13 9PT

robert.sansom@manchester.ac.uk

15.11.2019

Dear Prof. Hutchinson,

Thank you for your response regarding RSPB-2019-1596.R1 entitled "Bite marks and predation of fossil jawless fish during the rise of jawed vertebrates". We happy to hear that the associate editor and reviewers judge that we have addressed previous comments. In the revised version enclosed here we have addressed the remaining minor revisions, principally ensuring that our claims are not overstated and that other possible environmental factors behind extinction are highlighted.

Detailed response to the comments can be found below. The edited sections of the manuscript are highlighted in an uploaded revised version.

Your Sincerely

Robert Sansom and Emma Randle

Associate Editor Board Member: Comments to Author

The authors have revised their manuscript sufficiently well to address the reviewer's concerns. The language they use is more conservative now regarding their conclusions on predation-driven extinction, which I think is appropriate given the nature of their data (only 41 predated specimens). While I remain unconvinced of their hypothesis, I think this is a useful contribution to inspire debate and further research on the topic. Aside from the Reviewer's remaining comments and suggestions for changes (see attached PDF), I have a few more listed here:

The authors need to add verbiage in the Discussion regarding the validity of the other hypotheses, which they did not test. For example, environmental changes may well have contributed to the extinction of jawless fishes. This hypothesis could even be tested in a similar fashion to the biotic hypothesis tested here. I am not suggesting the authors do this, but merely that it could be done, and their analyses (as they stand) do not put a nail in the coffin of this explanation. Moreover, as the authors know, correlation does not equal causation, and this should be reiterated in the Discussion. Even if predation of jawless fishes by jawed fishes increased through time, it doesn't necessarily mean it caused the demise of jawless fishes. Verbiage of this nature could potentially go after line 255.

Response: We agree that environmental factors are likely an important component in the demise and extinction of ostracoderms. In the revised version we have revised our tone, and discussed this in more detail, including our previous paper which found correlations between the diversity of ostracoderms and relative sea-level changes:

“Contemporaneous with these biotic changes, ecosystems were also subject to dramatic abiotic changes. Rising sea-levels during the Devonian may have adversely affected ostracoderms due to their restriction to freshwater and shallow marine environments [1,9]. This environmental restriction, combined with their limited dispersal capability [27], has been invoked as a causative factor in the decline and eventual extinction of ostracoderms; generic diversity of ostracoderms has been demonstrated to be correlated with relative sea-level changes [1]. The increasing prevalence of bite marks through time identified here indicates that increasing pressure from predation by jawed vertebrates may have been an additional factor in ostracoderm decline and extinction. In both cases, correlations alone are not sufficient evidence of a causative factor, but in case of bite marks, we have direct evidence of a biotic interaction, the dynamics of which changed over time.”

*Lines 98-99: the methodology should be clarified here. **Response:** Details now provided.*

*Line 126: please remove the comma at end of this sentence. **Amended.***

*Line 145: need a 'with' here?. **Amended.***

*The Results section could benefit from some subheadings; the section from 158-183 could benefit from paragraph breaks. **Revised and Amended.***

Line 226: perhaps better worded as 'show a significant increase through time'
Response: Revised to “bite mark traces become increasingly prevalent through time”

*Line 244: phrasing here needs to be revised. **Revised and Amended.***

Referee: 1 Comments to the Author(s)

I thank the authors for addressing previous comments. I have provided some comments on the attached .pdf.

Response: We thank the reviewer for these comments. We have addressed the very minor revisions highlighted in the pdf, including listing the chi square degrees of freedom and using 'sub groups' rather than 'sub clades'.